# Dopamine neuronal loss contributes to memory and reward dysfunction in a model of Alzheimer's disease

Annalisa Nobili[1,2], Emanuele Claudio Latagliata[1], Maria Teresa Viscomi[1], Virve Cavallucci[1,2,†], Debora Cutuli[1], Giacomo Giacovazzo[1,3], Paraskevi Krashia[1,4], Francesca Romana Rizzo[1,4], Ramona Marino[2,5], Mauro Federici[1], Paola De Bartolo[1,6], Daniela Aversa[1,4], Maria Concetta Dell'Acqua[1,2], Alberto Cordella[1,4], Marco Sancandi[7], Flavio Keller[5], Laura Petrosini[1,7], Stefano Puglisi-Allegra[1,7], Nicola Biagio Mercuri[1,4], Roberto Coccurello[1,3], Nicola Berretta[1] & Marcello D'Amelio[1,2]

Alterations of the dopaminergic (DAergic) system are frequently reported in Alzheimer's disease (AD) patients and are commonly linked to cognitive and non-cognitive symptoms. However, the cause of DAergic system dysfunction in AD remains to be elucidated. We investigated alterations of the midbrain DAergic system in the Tg2576 mouse model of AD, overexpressing a mutated human amyloid precursor protein (APPswe). Here, we found an age-dependent DAergic neuron loss in the ventral tegmental area (VTA) at pre-plaque stages, although substantia nigra pars compacta (SNpc) DAergic neurons were intact. The selective VTA DAergic neuron degeneration results in lower DA outflow in the hippocampus and nucleus accumbens (NAc) shell. The progression of DAergic cell death correlates with impairments in CA1 synaptic plasticity, memory performance and food reward processing. We conclude that in this mouse model of AD, degeneration of VTA DAergic neurons at pre-plaque stages contributes to memory deficits and dysfunction of reward processing.

[1] Department of Experimental Neurosciences, IRCCS Santa Lucia Foundation, 00143 Rome, Italy. [2] Unit of Molecular Neurosciences, Department of Medicine, University Campus-Biomedico, 00128 Rome, Italy. [3] Institute of Cell Biology and Neurobiology (IBCN), National Research Council (CNR), 00143 Rome, Italy. [4] Department of Systems Medicine, University of Rome 'Tor Vergata', 00133 Rome, Italy. [5] Laboratory of Developmental Neuroscience and Neural Plasticity, Department of Medicine, University Campus-Biomedico, 00128 Rome, Italy. [6] Department of Technologies, Communication and Society (TECOS), University Guglielmo Marconi, 00193 Rome, Italy. [7] Department of Psychology, University Sapienza, 00185 Rome, Italy. † Present address: Institute of General Pathology, Università Cattolica School of Medicine, 00168 Rome, Italy. Correspondence and requests for materials should be addressed to M.D.A. (email: m.damelio@unicampus.it).

Alzheimer's disease (AD) is a neurological disorder characterized by cognitive and non-cognitive symptoms that are associated with brain atrophy[1,2]. The main histopathology accompanying AD involves the accumulation of neurofibrillary tangles and neuritic plaques, resulting in a progressive and massive neuronal loss that primarily affects the hippocampus and cortex[3–5].

The hippocampus is a critical brain structure for memory formation[6] and damage in this area is believed to be the principal cause for memory loss in AD patients. The hippocampal formation receives both cortical and sub-cortical inputs, the latter arriving mainly from the ventral tegmental area (VTA), locus coeruleus (LC), medial septal nucleus, raphe complex and the nucleus basalis of Meynert, all modulating hippocampal activity[7–11]. Pioneering neuropathological observations in post-mortem AD brain[12,13] demonstrated that the hippocampal formation and extrinsic (both cortical and sub-cortical) connections are disrupted at multiple levels, suggesting that the progressive structural alterations in the different brain areas may contribute to the worsening of memory and cognitive functions in AD patients.

Consistent with these observations, several alterations in the dopaminergic (DAergic) system have been reported in AD patients, including reduced levels of dopamine (DA) and its receptors[14–16]. One of the sources of DA in the hippocampus derives from DAergic neurons in the VTA[17]. DA is a well-recognized modulator of hippocampal synaptic plasticity and its binding to DAergic receptors in the dorsal hippocampus is a major determinant of memory encoding[18–20]. VTA DAergic neurons also target the nucleus accumbens (NAc) and cerebral cortex, mediating the control of incentive motivation and reward processing[17].

Owing to these functions of DA in the hippocampus and the mesolimbic system, we sought to establish how DAergic transmission in the hippocampus and NAc is affected in Tg2576 transgenic mice overexpressing the human APP695 protein with the 'Swedish' mutation (APPswe), which show behavioural and histopathological abnormalities that closely mimic early AD[21]. We hypothesized that damage of DAergic neurons could contribute to the deterioration of memory and to non-cognitive symptoms observed in these animals[22–25] and, plausibly, to early symptoms of AD patients[26]. We found a significant loss of tyrosine hydroxylase-positive (TH[+]) DAergic neurons in the VTA of Tg2576 mice, beginning at 3 months of age. Degeneration is selective for the VTA as DAergic neurons in the adjacent substantia nigra pars compacta (SNpc) were intact. Moreover, basal outflow of DA in the hippocampus and NAc shell is reduced, likely contributing to deficits in mesolimbic cognitive and non-cognitive symptoms. Notably, stimulation of the DAergic system by administration of the DA precursor levodopa (L-DOPA) or with selegiline, a monoamine oxidase-B inhibitor, completely rescues CA1 synaptic plasticity and dendritic spine density, and restores hippocampal post-synaptic density (PSD) composition, memory deficits and impairment in food reward processing. Our findings suggest a novel role for the VTA in the pathophysiology of AD-like symptoms at pre-plaque stages.

months of age (Fig. 1b). This reduction in TH[+] cell number was restricted to the VTA, as no difference was observed between Tg2576 and WT mice in the SNpc at any age tested (Fig. 1b). In contrast with TH[+] cells, the number of TH[−] cells remained constant (Fig. 1b), indicating that the decrease of TH[+] cells in the VTA of Tg2576 mice reflects a true loss of DAergic neurons and not a mere loss of TH immunoreactivity or down-regulation of TH expression. In agreement with the selective loss of VTA DAergic neurons, we detected more shrunken appearance and TUNEL-positive apoptotic cells in the VTA of 3-month-old Tg2576 mice, the age at which the TH[+] cell number starts to decline (Fig. 2a). To examine whether the neuronal apoptosis was associated with increased reactive astrocytes, we analysed the immunoreactivity of glial fibrillary acidic protein (GFAP), an astrocyte cell marker. Consistent with greater neuronal degeneration[27], we found a significant rise in the number of GFAP[+] cells in the VTA of 3-month-old Tg2576 mice, demonstrating an increase in reactive astrocytes in response to DAergic neuronal loss. Instead, GFAP immunoreactivity was unchanged in the SNpc of 3-month-old Tg2576 mice, in line with the observation that nigral DAergic neurons are intact (Fig. 2b).

We also examined the response of microglial cells using an ionized calcium binding adaptor molecule 1 (Iba1)-specific antibody[28]. Microglial cells in the healthy brain exist in a quiescent state (resting cells) with a relatively round cell body and long, thin processes, whereas in the initial state of activation (mildly activated) following brain damage, they proliferate rapidly, cell bodies enlarge and processes retract or become thicker[29,30]. In agreement with DAergic cell loss in the VTA, in 3-month-old Tg2576 animals we observed a change in microglial morphology and a significant increase in the ratio of mildly activated over resting cells, whereas microglia in the SNpc area remained unchanged (Fig. 2c). Thus, the selective degeneration in the VTA of Tg2576 mice is associated with apoptosis, increased reactive astrocytes and microglial activation.

In order to test whether the increase in neuronal degeneration in the VTA results from extracellular amyloid-β (Aβ)-plaque deposition, we examined the accumulation of Aβ deposits in midbrain DAergic neurons and in projection areas such as the hippocampus, dorsal striatum and NAc[31–33]. We found that extracellular Aβ-plaque deposition was absent in 6-month-old mice in all of the above-mentioned regions. However, we observed more intense cytoplasmic staining for APPswe in the hippocampus and NAc shell of Tg2576 mice compared to the other areas (Supplementary Fig. 1a). This was also confirmed by western blot analysis of full-length APPswe protein levels (Supplementary Fig. 1b). Importantly, DAergic neurons showed diffuse intracellular staining, and this was similar in the VTA and SNpc, suggesting the absence of intracellular Aβ accumulation in VTA neurons that could account for the selective cell degeneration (Supplementary Fig. 1a). Extracellular Aβ-plaque deposition was detected in the hippocampus of aging (11-month-old) Tg2576 mice, but not in the VTA (Supplementary Fig. 1c). These data indicate that the apoptotic cell death observed in VTA DAergic neurons as early as 3 months occurs at hippocampal pre-plaque stages in this animal model and does not appear to result from local Aβ-plaque deposition in the VTA.

## Results

### DAergic neuron degeneration in the VTA at pre-plaque stages.
We used TH immunostaining to identify DAergic neurons in Tg2576 and age-matched wild-type (WT) mice at different postnatal ages, and performed stereological cell counts in the VTA and SNpc of the midbrain (Fig. 1a). We observed a significant loss of DAergic neurons in Tg2576 mice, beginning at 3

### Reduced DA in the NAc shell and food reward deficits.
Midbrain DAergic neurons project to the striatum and NAc topographically along the mediolateral axis: the shell subregion of the NAc is innervated by medially located VTA neurons mediating motivation, reward-related cognition and multiple forms of memory; the NAc core is innervated by lateral VTA and medial SNpc neurons influencing motor responses related to

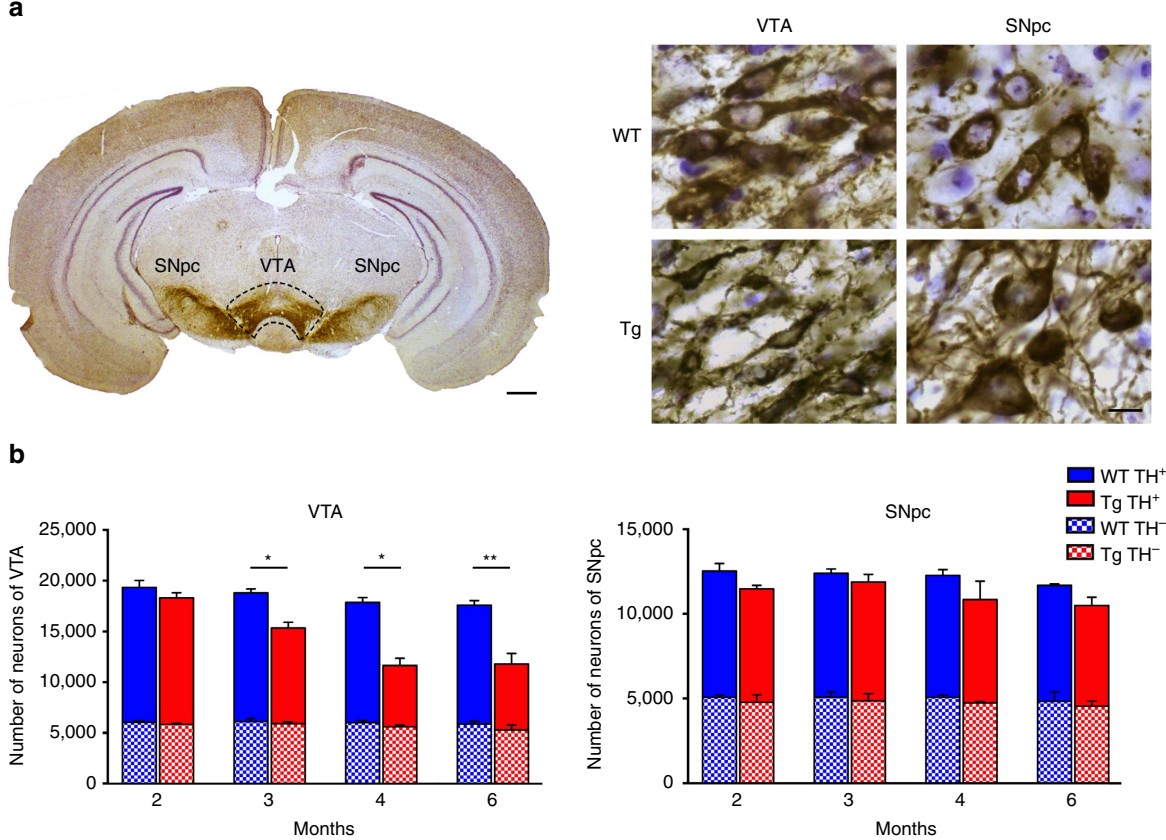

**Figure 1 | Tg2576 mice show selective loss of VTA DAergic neurons starting at 3 months of age.** (**a**) Coronal brain section from a 6-month-old WT mouse showing intense TH immunoreactivity (brown) in the VTA and SNpc. Sections were Nissl-counterstained (light blue). The dashed line indicates the anatomical boundaries separating the VTA from the SNpc (scale bar, 500 μm). On the right are higher magnification images (scale bar, 10 μm) showing TH$^+$ and Nissl-counterstained neurons in the VTA and SNpc of WT and Tg2576 (Tg) mice ($n = 7$ mice per genotype; 9 sections per animal). (**b**) The bar graphs show stereological quantification of TH$^+$ and TH$^-$ cell numbers in the VTA and SNpc in WT and Tg2576 mice at the indicated ages ($n = 7$ mice per genotype per age; 9 sections per animal). DAergic neuronal loss in Tg2576 mice is selective for the VTA with the onset at 3 months of age (two-tailed unpaired t-test: 3 and 4 months, *$P = 0.003$; 6 months, **$P = 0.002$). Data represent mean ± s.e.m.

reward stimuli, whereas the dorsal striatum is almost exclusively innervated by laterally located SNpc neurons controlling voluntary movement[32,34,35]. To determine whether the selective loss of DAergic neurons affects the release of DA in projecting areas that are associated with the VTA, but not with the SNpc, we first measured DA outflow in the NAc shell and core and in the dorsal striatum, using amperometric recordings of DA release in acute brain slices (Fig. 3a). In 6-month-old Tg2576 mice, when the DAergic neuronal loss in the VTA reaches a plateau (Fig. 1a), the evoked DA was significantly decreased in the NAc shell (Fig. 3b). Instead, no difference was observed in the NAc core or in the striatum (Fig. 3b). In addition, in 2-month-old mice, when the number of DAergic neurons in the Tg2576 VTA is still unchanged compared to WT animals, the release of DA in the NAc shell is similar between genotypes (Supplementary Fig. 2a). These data show that in Tg2576 mice DA outflow in the NAc shell is normal before the onset of DAergic cell death and suggest that its reduction in the NAc shell at 6 months results from VTA DAergic neuronal loss. Consistent with the amperometric results, staining for the DA transporter (DAT), used as a specific marker of DAergic terminals, was reduced in the NAc shell, but not in the NAc core of 6-month-old Tg2576 mice (Fig. 3c).

Microdialysis experiments performed on freely moving animals confirmed the reduced outflow of DA in the NAc shell of 6-month-old Tg2576 mice (Fig. 3d). Surprisingly, *in vivo* microdialysis detected significantly reduced levels of basal DA

also in the striatum (Fig. 3e). Considering that both the SNpc DAergic neurons (Fig. 1b) and striatal DA release by phasic DAergic activity (Fig. 3b) were intact in Tg2576 mice, it is likely that the reduced basal levels of DA in the striatum might reflect alterations in the control of tonic DA release[36,37].

Due to the importance of the mesolimbic system for reward and motivation processing[33,38], we examined whether the reduced outflow of DA from the VTA to the NAc shell could be associated with dysfunctional mesolimbic reward-associated processing in Tg2576 mice. To this end, we measured the responses of 6-month-old mice in a chocolate-elicited conditioned place preference (CPP) task (Supplementary Fig. 2b). During the pre-conditioning session, mice were left to freely explore a two-chamber apparatus. Irrespective of genotype, all animals spent equal amounts of time in both chambers, showing no preference for either compartment (Supplementary Fig. 2c). Exposure to palatable food in WT mice resulted in a significant increase in the time spent in the chocolate-paired chamber during the testing phase (Fig. 3f). In contrast, place conditioning was absent in Tg2576 mice: these animals spent equal amounts of time in the chocolate-paired chamber during the pre-conditioning and testing session and failed to develop any preference for the chamber with the rewarding food (Fig. 3f). In addition, Tg2576 mice consumed less chocolate during the conditioning phase (Fig. 3g). The reduction in place preference during palatable food-seeking, together with decreased chocolate

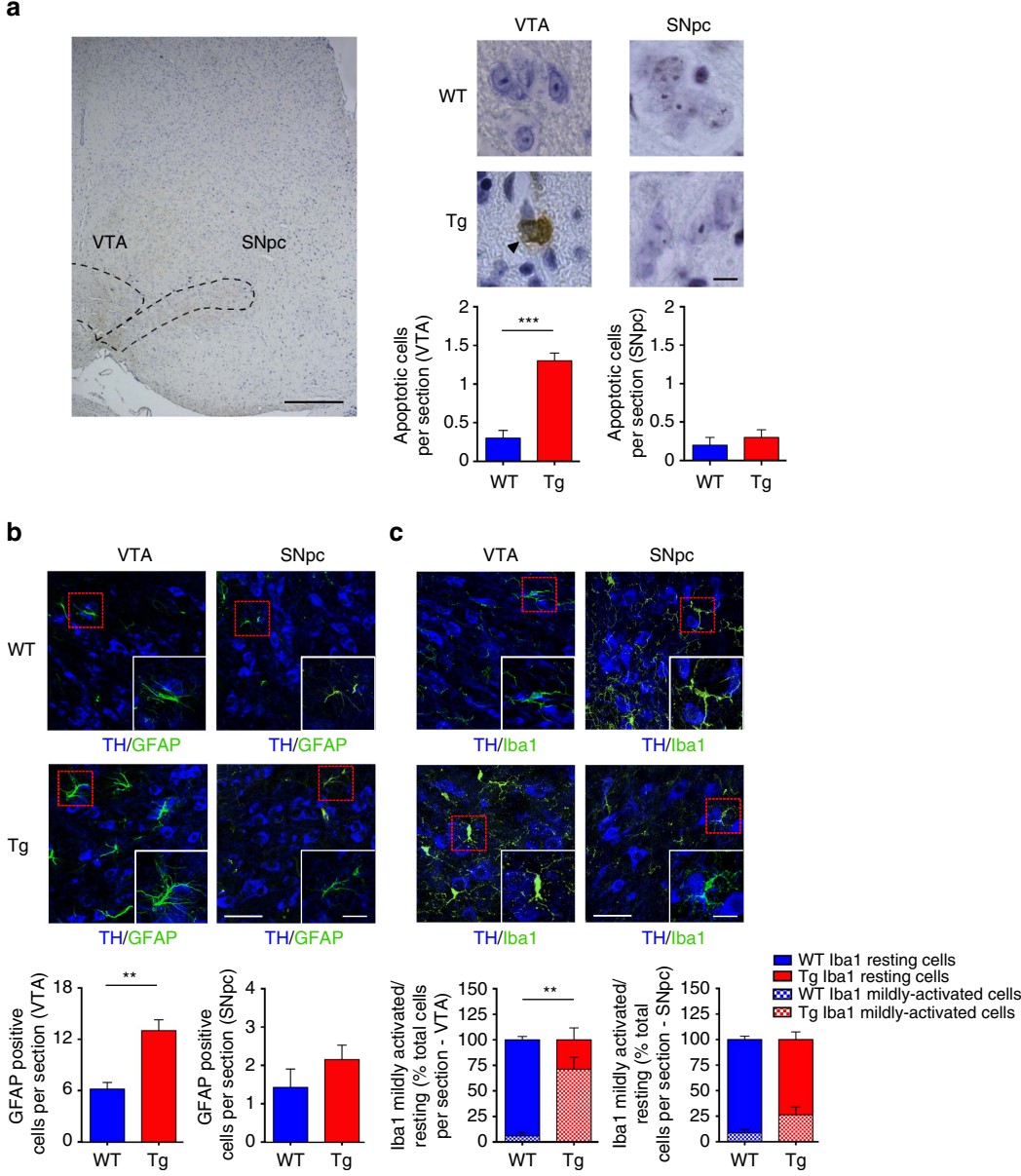

**Figure 2 | VTA DAergic neuronal death in Tg2576 mice at pre-plaque stage is associated with apoptosis and glia inflammation. (a)** Coronal brain hemi-section delineating VTA and SNpc areas (dashed lines; scale bar, 250 μm) and representative photomicrographs of TUNEL-positive neurons from 3-month-old WT and Tg2576 mice (scale bar, 10 μm). The intense dark-brown staining indicates apoptotic cells (arrowhead). Sections were Nissl-counterstained (light blue). The bar graphs represent the average number of TUNEL-positive apoptotic neurons per section in the analysed areas ($n = 7$ mice per genotype, 9 sections per animal; two-tailed unpaired $t$-test ***$P < 1.00 \times 10^{-4}$). **(b)** Analysis of confocal Z-stack double-labelling of TH- and GFAP- immunostaining in brain sections containing the VTA and SNpc from 3-month-old mice (scale bar, 50 μm). Insets show individual GFAP-positive cells at higher magnification (scale bar, 20 μm). The bar plots represent the mean number of GFAP-positive cells per section in the indicated areas ($n = 7$ mice per genotype, 9 sections per animal; two-tailed unpaired $t$-test **$P = 0.002$). **(c)** Double-labelling for TH and Iba1 in brain sections containing the VTA and SNpc from 3-month-old mice (scale bar, 50 μm). The insets show examples of resting microglia in WT mice and in the SNpc of Tg2576 mice, characterized by round cell bodies and long processes, and a mildly activated cell in the VTA of Tg2576 mice with more intense fluorescence, enlarged cell body and retracted processes (scale bar, 20 μm). Note also the increased proliferation of Iba1-positive cells in the Tg2576 VTA. The bar plots represent the mean number of Iba1-positive resting and mildly activated cells shown as percentage of the total number of Iba1-positive cells per section ($n = 4$ mice per genotype, 9 sections per animal; for the ratio of resting/mildly activated cells: two-tailed unpaired $t$-test **$P = 0.003$). Data are mean ± s.e.m.

ingestion, are consistent with depletion of DA from the NAc shell where DA governs reward and effort-related decision-making[33,38] and argue for deficits in reward-associated cognition and for depressive-like symptoms in 6-month-old Tg2576 mice. In 2-month-old Tg2576 mice, when the outflow of DA in the NAc shell is still normal, there is no impairment in place preference or food consumption (Supplementary Fig. 2d,e).

**Reduced DA in the hippocampus and memory impairments.** The DAergic projections from the VTA to the hippocampus form the upward arm of a functional loop that controls novel memory formation[31], with DA playing an important role in modulating hippocampal synaptic plasticity and function[18–20,39]. In light of the degeneration of the VTA DAergic neurons in Tg2576 mice, we investigated a possible depletion of DA outflow in the

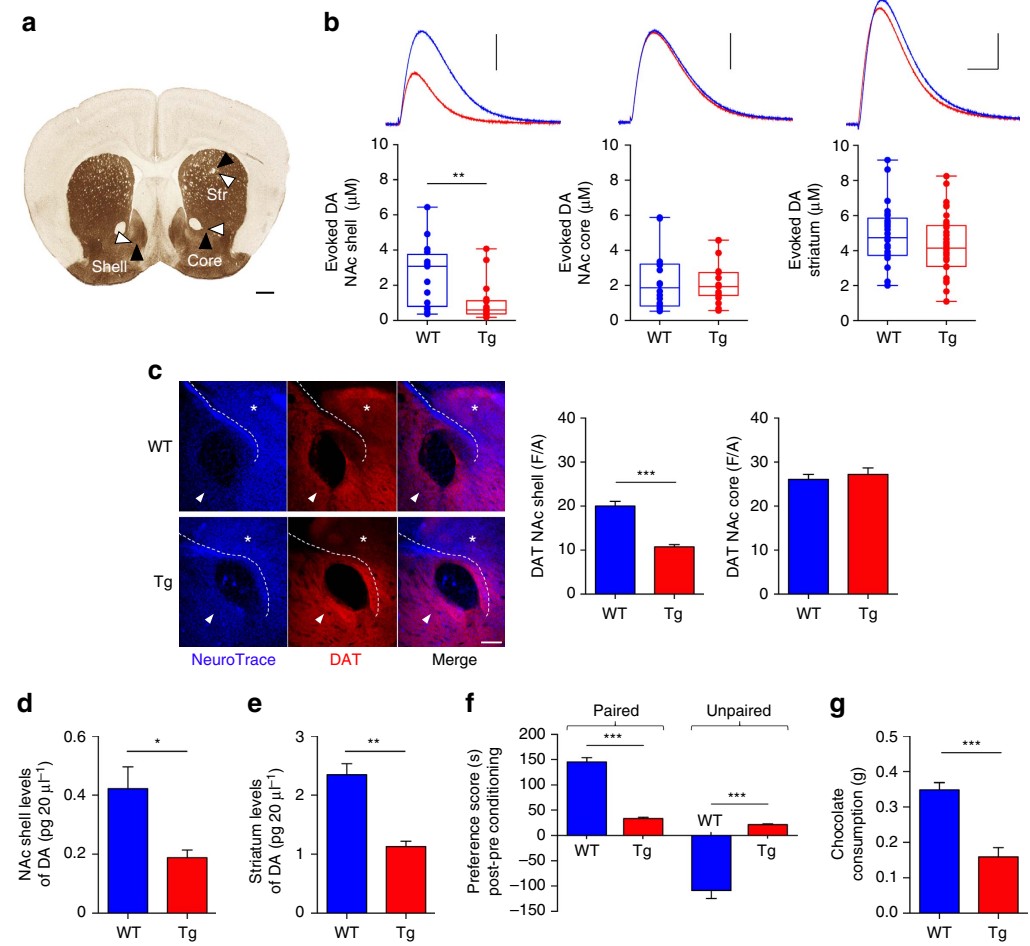

**Figure 3 | Reduced DA outflow in the NAc shell and deficits in mesolimbic reward processing in 6-month-old Tg2576 mice.** (**a**) Summary of stimulating (black arrowheads) and recording electrode (white arrowheads) placements during amperometric measurements of evoked DA in the NAc (Shell, Core) and striatum (Str; scale bar, 500 μm). (**b**) Evoked DA concentration in the indicated areas ($n = 16$–25 slices from 6–8 WT mice, 18–43 slices from 6–10 Tg2576 mice) and example traces from 6-month-old WT and Tg2576 mice (vertical scale bars: NAc shell and core, 50 pA; striatum, 100 pA; horizontal scale bar, 250 ms) recorded with a carbon fibre electrode of equal calibration (two-tailed unpaired $t$-test **$P = 0.004$). In this and other figures, in box-and-whisker plots the centre lines denote medians, edges represent upper and lower quartiles and whiskers show minimum and maximum values. Points are individual experiments. (**c**) Z-stack double immunofluorescent labelling for NeuroTrace and DAT in NAc coronal sections showing the NAc shell (asterisk) and core (arrowhead; scale bar, 200 μm). Bar plots show densitometric values of DAT levels in 6-month-old mice ($n = 3$ per genotype, 4 sections per animal; two-tailed unpaired $t$-test ***$P = 1.00 \times 10^{-7}$). (**d**,**e**) Microdialysis measurements of DA outflow in the NAc shell (**d**) and dorsal striatum (**e**) in 6-month-old mice (**d**: $n = 6$ WT and 5 Tg2576; one-way ANOVA: $F_{1,9} = 5.138$, *$P = 0.049$; **e**: $n = 4$ WT and 5 Tg2576; one-way ANOVA: $F_{1,7} = 13.067$, **$P = 0.009$). (**f**) Chocolate-induced place preference in 6-month-old mice ($n = 5$ mice per genotype) showing average time spent in paired and unpaired chambers in post-conditioning session, minus the time spent in the same chambers during the pre-conditioning session of a CPP test (two-way repeated measures ANOVA: chamber, $F_{1,8} = 280.76$, $P < 1.00 \times 10^{-4}$; chamber × genotype, $F_{1,8} = 231.34$, $P < 1.00 \times 10^{-4}$; genotype, $F_{1,8} = 0.84$, $P = 0.380$; ***$P < 1.00 \times 10^{-4}$ with Tukey's *post hoc* test). (**g**) Chocolate consumption during CPP conditioning sessions ($n = 5$ mice per genotype; two-tailed unpaired $t$-test ***$P < 1.00 \times 10^{-4}$). Data in **c**–**g** represent mean ± s.e.m.

hippocampus and whether this could be correlated with memory performance impairments in these mice.

Because DAergic innervation of the hippocampus is sparser than that of the NAc, amperometric recordings of DA release in acute hippocampal brain slices was not feasible. However, microdialysis experiments on freely moving animals indicated a reduced DA outflow in the hippocampus of 6-month-old Tg2576 mice compared to age-matched controls (Fig. 4a). On the other hand, noradrenaline outflow in the hippocampus did not differ between genotypes (Fig. 4b), suggesting that at this age the noradrenergic projection from the LC is preserved. Considering that LC TH$^+$ neurons can also co-release DA together with noradrenaline[40,41] and that the majority of TH$^+$ projections in the hippocampus originate from the LC[41,42], we performed a stereological cell-count of TH$^+$ neurons in the LC of 6-month-old

Tg2576 mice. LC stereological cell-count confirmed that, at this age, the number of TH$^+$ neurons in this brain region did not differ between genotypes (Fig. 4c,d). These data suggest that the reduced DA outflow in the hippocampus of 6-month-old Tg2576 mice is principally due to the established VTA DAergic neuron degeneration, although reduced axonal DA release from LC TH$^+$ neurons cannot be excluded, especially in older animals[43,44]. In line with these data, we found that both the TH protein levels (Fig. 4e, Supplementary Fig. 6a) and TH immunoreactivity (Fig. 4f) in the hippocampus were significantly reduced in 6-month-old Tg2576 mice. Instead, TH protein levels of Tg2576 were identical to WT animals in the striatum (Fig. 4g, Supplementary Fig. 6b), in accordance with amperometric data in this area and the observation that SNpc DAergic neurons, mainly projecting to the striatum, are still intact at this age.

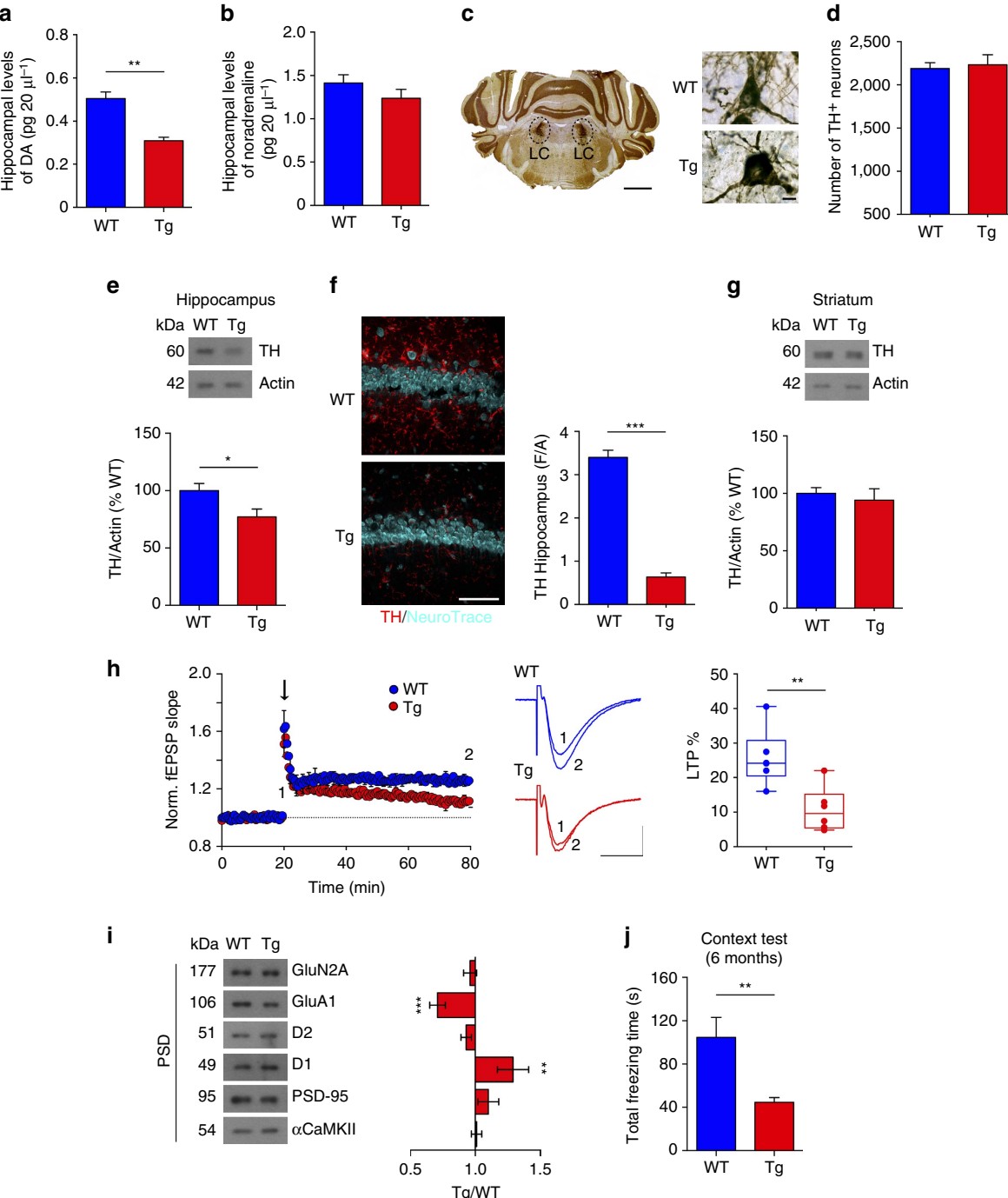

**Figure 4 | Six-month-old Tg2576 mice show reduced DA outflow in the hippocampus and synaptic plasticity and memory deficits.** (**a**,**b**) *In vivo* microdialysis for DA (**a**) and noradrenaline (**b**) in the hippocampus of 6-month-old WT ($n=6$) and Tg2576 ($n=5$) mice (**a**: $F_{1,9}=5.138$, **$P=0.009$; (**b**): $F_{1,9}=0.688$, $P=0.428$, one-way ANOVA). (**c**) TH immunoreactivity (brown) in the LC (scale bar, 500 μm) of a 6-month-old WT mouse and images of TH$^+$ neurons in WT and Tg2576 mice (scale bar, 10 μm). (**d**) Quantification of LC TH$^+$ neurons in 6-month-old mice ($n=6$ per genotype; 9 sections per animal). (**e**) Immunoblots of total hippocampal TH protein from 6-month-old mice ($n=6$ per genotype) and densitometric quantification of changes in grey values (two-tailed unpaired *t*-test *$P=0.037$). (**f**) TH/NeuroTrace double-labelling in CA1 sections (scale bar, 50 μm) and TH densitometric levels in 6-month-old mice ($n=3$ per genotype, 4 sections per animal; two-tailed unpaired *t*-test ***$P=5.00\times10^{-5}$). (**g**) As in **e** showing total TH protein from the dorsal striatum ($n=6$ mice per genotype). (**h**) Normalized CA3-to-CA1 fEPSP mean slope (± s.e.m. every 2 min) recorded from the CA1 dendritic region in slices from 6-month-old mice. A high frequency conditioning train was delivered (arrow) following a 20 min baseline. Traces (scale bars, 100 μV, 10 ms) are fEPSPs recorded during baseline (1) and 1 h after the train (2). The plot indicates the degree of potentiation at 55–60 min after the train (WT: $n=6$ slices from 4 mice; Tg2576: $n=6$ slices from 5 mice; two-tailed unpaired *t*-test **$P=0.006$). (**i**) Immunoblots of hippocampal PSD proteins from 6-month-old mice ($n=8$ per genotype), probed with the indicated antibodies, and densitometric quantification of changes in grey values expressed as mean ratio of Tg/WT (two-tailed unpaired *t*-test: D1, **$P=0.002$; GluA1, ***$P=0.001$). (**j**) Total freezing time during the CFC context test (6 mice per genotype; two-tailed unpaired *t*-test **$P=0.009$). Except from the box-and-whisker plot in **h**, values represent mean ± s.e.m.

Earlier studies describe structural alterations in hippocampal synapses and impairments in synaptic plasticity in Tg2576 mice that may underlie some of the cognitive deficits in these animals[22–24]. We and others have previously shown that in 3-month-old animals, at the age of onset of DAergic cell death, long-term potentiation (LTP) in CA3-to-CA1 synapses of Tg2576 mice is unaffected[24,25] while PSD composition changes already occur, with reduced levels of the AMPAR GluA1 subunit, decreased density of CA1 pyramidal neuron dendritic spines and altered glutamatergic transmission[25]. Here, in order to investigate whether synaptic impairments in the Tg2576 hippocampus parallel the advancement of VTA DAergic cell loss, we measured LTP in CA3-to-CA1 synapses in 2- and 6-month-old Tg2576 mice. In 2-month-old animals, when DAergic neuron numbers in the VTA are still intact, LTP in Tg2576 animals did not differ significantly from age-matched controls (Supplementary Fig. 3a). Instead, LTP magnitude was

significantly lower in 6-month-old Tg2576 mice (Fig. 4h), an age at which VTA DAergic cell death is pronounced and DA outflow in the hippocampus is significantly reduced. Thus, in Tg2576 mice potentiation at CA3-to-CA1 synapses deteriorates with age and this impairment appears to follow the advancement of VTA DAergic cell death. In contrast to LTP, short-term potentiation was unchanged in 6-month-old Tg2576 mice as the paired-pulse ratio (PPR) was similar between genotypes at the different interpulse intervals tested (Supplementary Fig. 3b). The input–output relationship was also similar between WT and Tg2576 mice, indicating no change in basal synaptic transmission (Supplementary Fig. 3c).

As PSD modifications precede functional alterations of synapses[25], we analysed protein expression in the hippocampal PSD fraction in 6-month-old Tg2576 mice. Consistent with our previous observations in 3-month-old animals[25], the surface expression of the AMPAR GluA1 subunit was down-regulated in

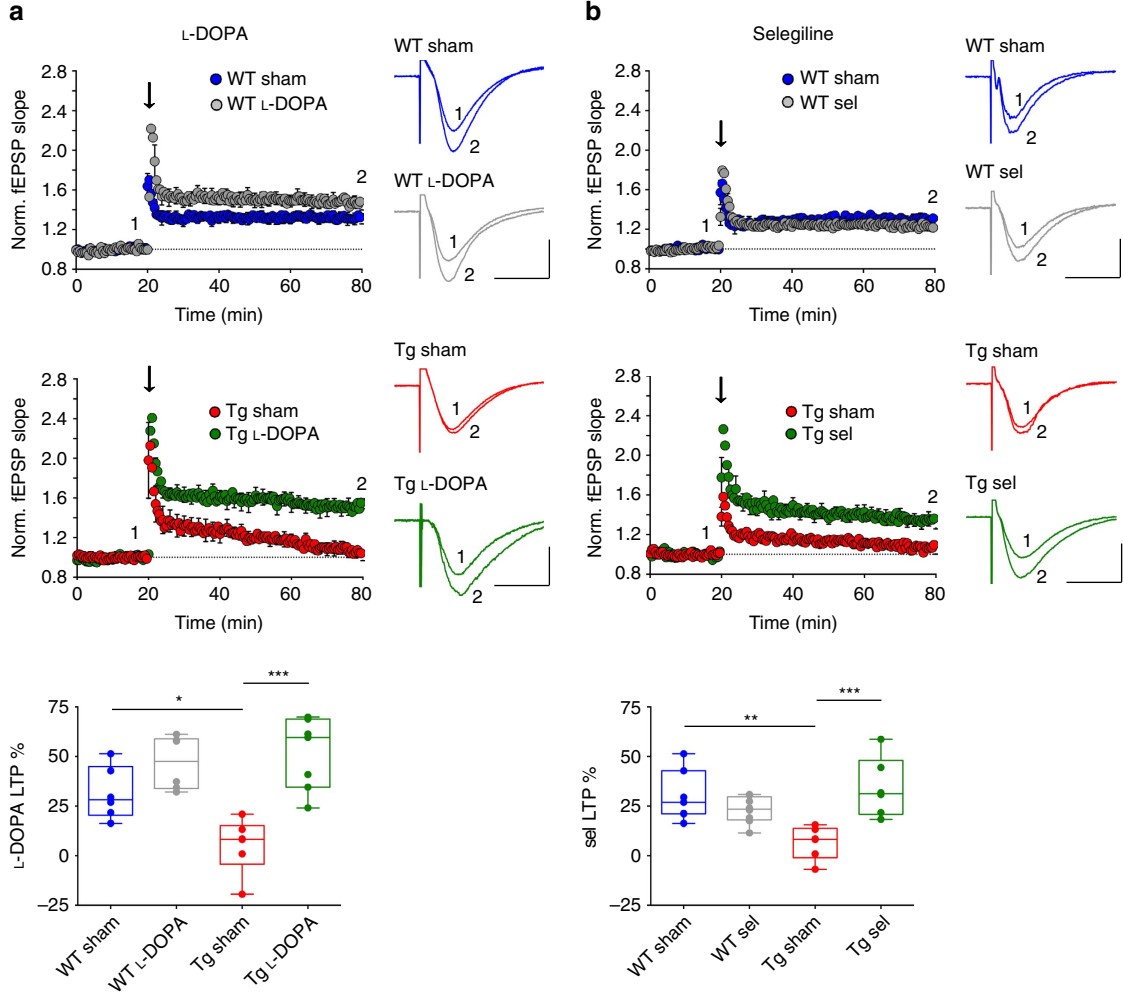

**Figure 5 | Sub-chronic L-DOPA or selegiline treatment rescues CA3-to-CA1 plasticity deficits in 6-month-old Tg2576 mice. (a,b)** Running plots show normalized fEPSP mean slope ( ± s.e.m. displayed every 2 min) recorded from the dendritic region of CA1 neurons in hippocampal slices from 6-month-old saline-treated (sham) and L-DOPA- (**a**) or selegiline (sel)-treated (**b**) WT and Tg2576 mice. The arrows indicate when a high frequency train was delivered by stimulating the Schaffer collateral pathway in the CA3 region. Traces are superimposed fEPSPs recorded during baseline (1) and 1 h after the train (2). The box-and-whisker plots below indicate the degree of potentiation, measured as fEPSP slope increase from baseline, 55–60 min after the train (**a**, WT: $n = 6$ slices from 3 sham, 6 slices from 4 L-DOPA-treated mice; Tg: $n = 6$ slices from 3 sham, 7 slices from 3 L-DOPA-treated mice; two-way ANOVA: genotype × treatment, $F_{1,21} = 6.51$, $P = 0.019$; genotype, $F_{1,21} = 3.22$, $P = 0.087$; treatment, $F_{1,21} = 26.09$, $P < 1.00 \times 10^{-4}$; WT sham-Tg sham *$P < 0.050$, Tg sham-Tg L-DOPA ***$P < 0.001$ with Bonferroni's *post hoc* test. (**b**) WT: $n = 7$ slices from 3 sham mice, 8 slices from 4 sel-treated mice; Tg2576: $n = 6$ slices from 3 sham, 6 slices from 4 sel-treated mice; two-way ANOVA: genotype × treatment, $F_{1,23} = 16.61$, $P = 5.00 \times 10^{-4}$; genotype, $F_{1,23} = 2.00$, $P = 0.171$; treatment, $F_{1,23} = 5.99$, $P = 0.022$; WT sham-Tg sham **$P < 0.010$, Tg sham-Tg sel ***$P < 0.001$ with Bonferroni's *post hoc* test. Both L-DOPA and selegiline increase LTP in Tg2576 mice while having no effect on WT animals (scale bars for traces: 100 μV, 10 ms).

6-month-old Tg2576 mice (Fig. 4i, Supplementary Figs 3d,e and 6c). Conversely, we found that the hippocampal levels of D1 receptors increased significantly in the PSD of Tg2576 mice, suggesting the development of a compensatory response to low levels of basal DA. We did not observe differences in other post-synaptic components, including D2 receptors (Fig. 4i).

Finally, we examined the development of memory deficits in a contextual fear-conditioning (CFC) task. During CFC, the total freezing time after re-exposure to an aversive context declined in 6-month-old Tg2576 mice (Fig. 4j, Supplementary Fig. 3f). Importantly, as previously reported[23,25], memory deficits were absent in 2-month-old Tg2576 mice (Supplementary Fig. 3g), in line with the absence of DAergic cell death in the VTA before 3 months of age.

**L-DOPA and selegiline restore memory and reward deficits.** Previous studies have shown that pharmacological manipulations aimed at increasing the DAergic transmission in the hippocampus and cortex could improve synaptic functions, cognitive impairments and memory deficits in AD patients[45–48] and AD-like experimental models[49–53]. Prompted by the finding that DAergic cell death progression in the VTA occurs around the time of worsening of synaptic dysfunction and memory deficits in the Tg2576 hippocampus, we hypothesized that the enhancement of the DAergic tone could improve the hippocampal alterations. To this aim we examined the effects of sub-chronic treatment with the DA precursor L-DOPA on hippocampal synaptic plasticity (see Supplementary Fig. 4a for drug administration protocol). Six-month-old WT animals sub-chronically treated with L-DOPA had a slight, yet not significant increase in the magnitude of CA3-to-CA1 LTP compared to saline-treated (sham) controls (Fig. 5a). On the other hand, L-DOPA treatment rescued LTP deficits in Tg2576 mice by restoring LTP to the levels of WT animals (Fig. 5a). The rescue effect of L-DOPA in Tg2576 mice was reproduced by acute, bath-application of L-DOPA (10 μM) during LTP recordings and it was mimicked by the selective D1/D5 receptor agonist SKF38393 (10 μM; Supplementary Fig. 4b). On the other hand, quinpirole (100 nM), a selective D2 receptor agonist, failed to change LTP in slices from Tg2576 mice, suggesting that the rescue of LTP following L-DOPA treatment and increased DA availability is mediated by D1/D5 receptors.

To confirm these data and examine the long-term effects of the increased availability of endogenous DA, we treated 6-month-old animals with selegiline, an irreversible selective inhibitor of monoamine oxidase-B (ref. 54; Supplementary Fig. 4c). Microdialysis experiments on freely moving animals confirmed that sub-chronic treatment with selegiline increased DA outflow in the hippocampus of Tg2576 mice (Supplementary Fig. 4d). Similarly to L-DOPA, selegiline treatment rescued the LTP magnitude in 6-month-old Tg2576 mice, without affecting WT animals (Fig. 5b). Thus, both exogenously (L-DOPA) and endogenously (selegiline)-induced increases in DA availability improved hippocampal plasticity, supporting our hypothesis that the reduced DA outflow in the AD hippocampus worsens synaptic dysfunctions in Tg2576 mice.

As expected, selegiline failed to rescue TH levels in 6-month-old Tg2576 mice (Fig. 6a, Supplementary Fig. 7a). However, the selegiline-induced increase in DA availability restored GluA1 phosphorylation (Ser845) levels in Tg2576 mice while having no effect in controls (Fig. 6a). This activity appeared to depend on the effects of selegiline on D1 exposure because it triggered the removal of D1 receptors from the PSD in both WT and Tg2576 mice (Fig. 6b, Supplementary Fig. 7b), likely as compensatory response to DA availability. The activation of D1 receptors was

followed by the increase of GluA1 expression in the PSD (Fig. 6b), in line with previous reports[55,56]. Similar results were also obtained by L-DOPA sub-chronic treatment that led to a restoration of both D1 receptors and GluA1 expression in hippocampal PSD fractions (Fig. 6c, Supplementary Fig. 7c). The recovery of the hippocampal CA3-to-CA1 synaptic plasticity and PSD composition in Tg2576 mice by selegiline was also correlated with the restoration of spine density in pyramidal neuron apical dendrites located in the CA1 stratum radiatum (Fig. 6d,e).

In view of the fact that treatments with L-DOPA or selegiline improve hippocampal synaptic plasticity and PSD composition, we then asked whether the impaired memory function in Tg2576 mice could be ameliorated with treatment. Indeed, both selegiline (Supplementary Fig. 5a) and L-DOPA (Supplementary Fig. 4a) treatments increased the total freezing time elicited by re-exposure to aversive stimuli during a CFC task in 6-month-old Tg2576 mice (Fig. 7a,b, Supplementary Fig. 5b,c). Furthermore, selegiline improved spatial memory of Tg2576 mice, as indicated by the increased distance travelled in the target quadrant that previously contained the escape platform during the Probe phase of a Morris water maze test (Fig. 7c, Supplementary Fig. 5d–f).

Finally, we investigated whether the selegiline treatment could restore the defects in CPP and food consumption observed in Tg2576 mice during the CPP test (Supplementary Fig. 5g). As expected, although saline-treated Tg2576 animals were unable to show increased preference for the chamber associated with the rewarding food and consumed less chocolate during the conditioning phase, these deficits were absent from selegiline-treated mice (Fig. 7d,e). We conclude that the increased availability of DA in the hippocampus and NAc can ameliorate mnesic deficits and impairment in mesolimbic reward processing in 6-month-old Tg2576 mice.

## Discussion

In recent years accumulating evidence has demonstrated a link between AD-related memory dysfunction and deficits in DA signalling in patients and experimental models of AD (refs 49–52,56–61). Here, in a mouse model of AD, at a stage when no Aβ-plaque deposition, hyperphosphorylated tau tangles or any sign of neuronal loss in cortical and hippocampal regions involved in memory deficits has yet occurred[21,23,24], we provide evidence that a specific apoptotic process is taking place in the VTA, causing a progressive degeneration of the DAergic neuronal population. This neuronal loss is selective for the VTA DAergic neurons, as demonstrated by the finding that the number of TH[−] neurons remains substantially unchanged.

The loss of VTA DAergic neurons is paralleled by a reduced outflow of DA in the hippocampus and in the shell region of the NAc. Curiously, DA outflow to the NAc core (and DAT levels) was unchanged in Tg2576 mice, even though this region is heavily innervated by VTA DAergic neurons[34]. However, it should be noted that the NAc shell and core are innervated by anatomically distinct subpopulations of highly heterogeneous VTA neurons regarding their properties[34,62,63], with the majority of shell-projecting neurons located in the paranigral nucleus in the medial VTA and the majority of core-projecting neurons located in the parabrachial pigmented area of the lateral VTA[34,35], suggesting that medial VTA DAergic neurons may show more vulnerability in Tg2576 mice.

The reduced DA outflow in the Tg2576 hippocampus and NAc shell, brain areas primarily implicated in memory and reward, respectively, might largely contribute to the deficits of hippocampus-dependent memory and synaptic plasticity, as well as impairment in reward processing. We found that the progression of these deficits in the Tg2576 mouse model follows

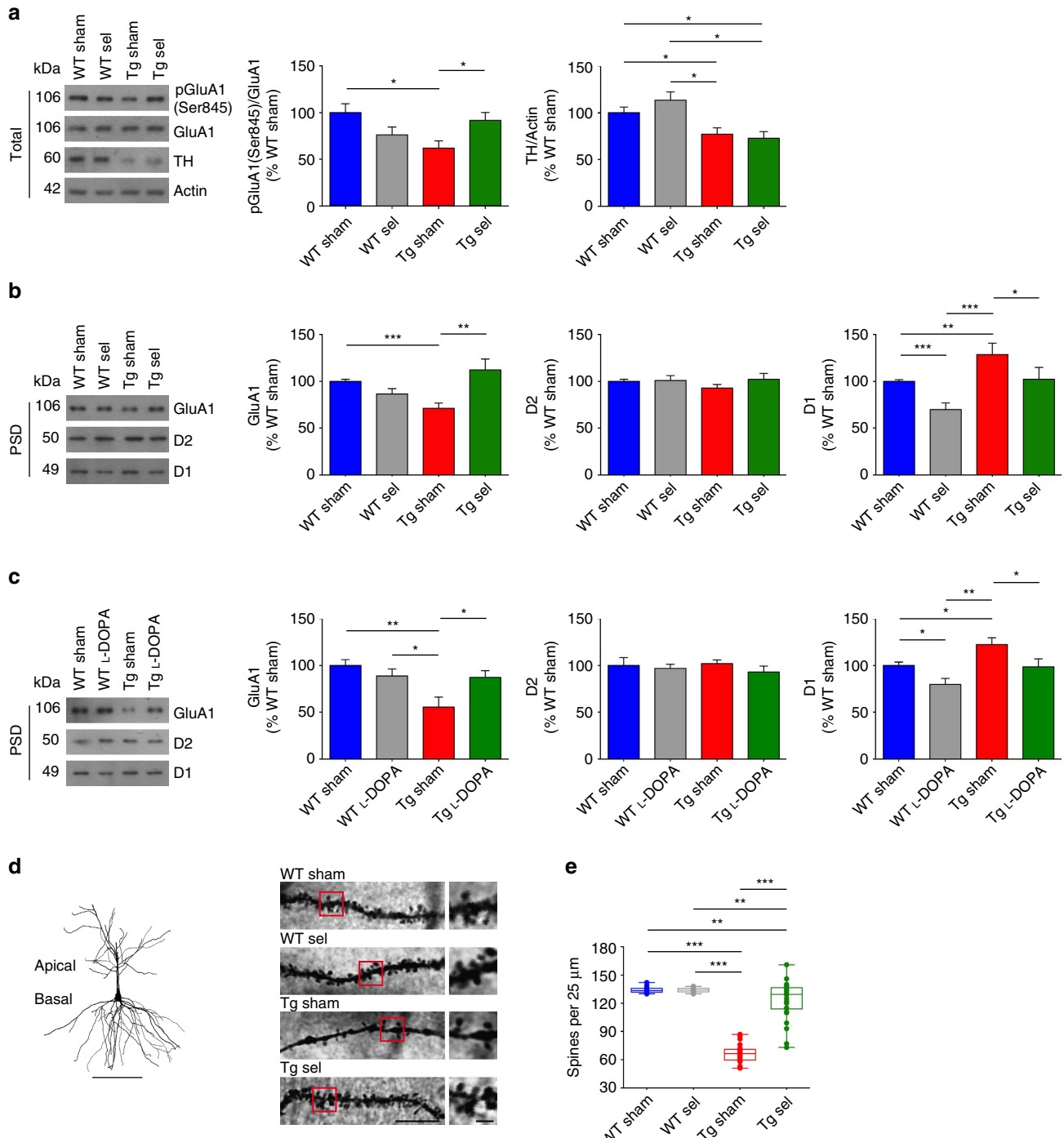

**Figure 6 | Sub-chronic selegiline or L-DOPA treatment rescues hippocampal PSD composition and dendritic spine density in Tg2576 mice.** (a–c) Representative immunoblots of total (**a**) and PSD (**b**) hippocampal proteins from 6-month-old WT and Tg2576 mice treated with selegiline (sel) or saline (sham) and PSD proteins (**c**) prepared from L-DOPA or saline-treated mice, probed with the indicated antibodies, and densitometric quantification of changes in grey values. For total proteins actin was used as loading control. Selegiline does not rescue TH protein levels but selegiline and L-DOPA restore PSD composition in Tg2576 mice (**a**: 8 mice per group; two-tailed unpaired $t$-test for pGluA1/GluA1: WT sham-Tg sham *$P = 0.012$, Tg sham-Tg sel *$P = 0.027$; for TH/actin: WT sham-Tg sham *$P = 0.029$, WT sham-Tg sel *$P = 0.022$, WT sel-Tg sham *$P = 0.017$, WT sel-Tg sel *$P = 0.015$; (**b**) 10 mice per group; two-tailed unpaired $t$-test for GluA1: WT sham-Tg sham ***$P = 8.00 \times 10^{-4}$, Tg sham-Tg sel **$P = 0.010$; for D1: WT sham-WT sel ***$P = 2.00 \times 10^{-4}$, WT sham-Tg sham **$P = 0.009$, WT sel-Tg sham ***$P = 3.00 \times 10^{-4}$, Tg sham-Tg sel *$P = 0.028$; (**c**) 8 mice per group; two-tailed unpaired $t$-test for GluA1: WT sham-Tg sham **$P = 0.008$, WT L-DOPA-Tg sham *$P = 0.031$, Tg sham-Tg L-DOPA *$P = 0.032$; for D1: WT sham-WT L-DOPA *$P = 0.020$, WT sham-Tg sham *$P = 0.028$, WT L-DOPA-Tg sham **$P = 0.002$, Tg sham-Tg L-DOPA *$P = 0.039$). (**d**) Reconstruction of a WT sham Golgi-stained CA1 pyramidal neuron (scale bar, 100 μm) and representative segments of apical dendrites from 6-month-old sel- and sham-treated mice (scale bar, 10 μm). Insets show high-magnification micrographs (scale bar, 2 μm). (**e**) Spine density (mean spine number per 25 μm dendrite segment) is increased in Tg2576 animals after selegiline treatment (**d,e**: $n = 3$ mice per group, 10 pyramidal neurons per mouse; two-tailed unpaired $t$-test: WT sham-Tg sham ***$P < 1.00 \times 10^{-4}$, WT sham-Tg sel **$P = 0.010$, WT sel-Tg sham ***$P < 1.00 \times 10^{-4}$, WT sel-Tg sel **$P = 0.010$, Tg sham-Tg sel ***$P < 1.00 \times 10^{-4}$). Data represent mean ± s.e.m.

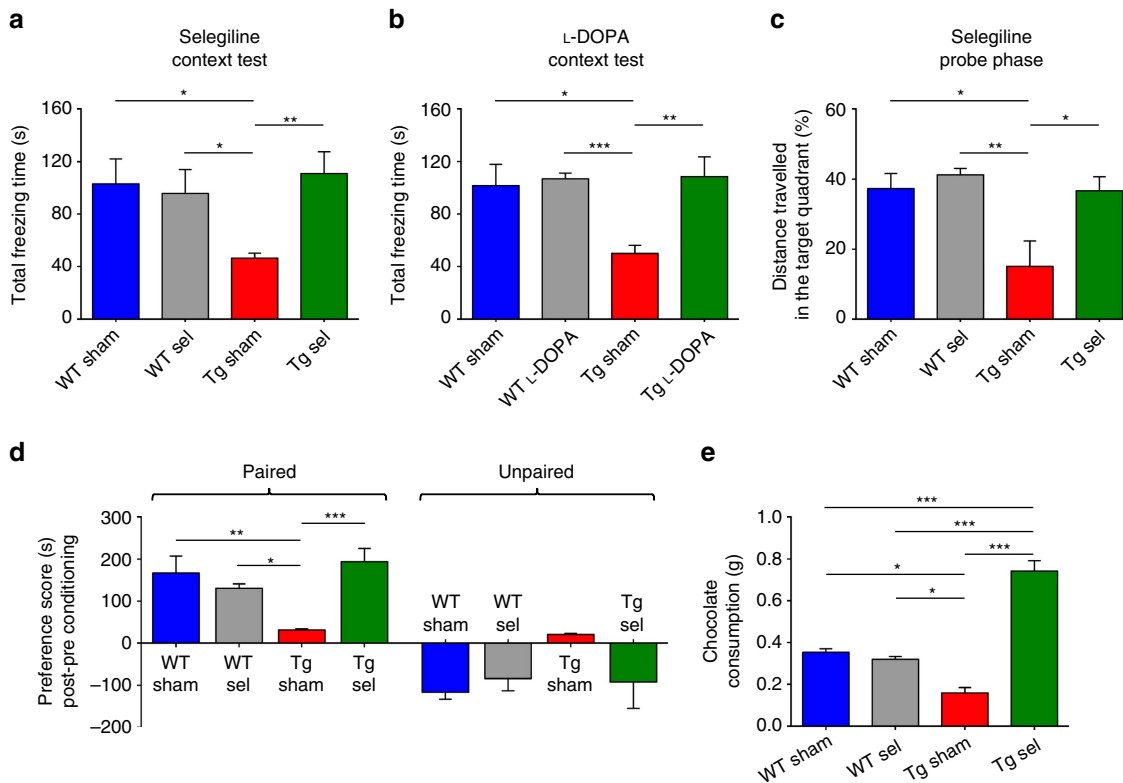

**Figure 7 | Sub-chronic selegiline or L-DOPA treatment rescues memory performance and reward processing in Tg2576 mice.** (**a,b**) Total freezing time during the CFC context test in 6-month-old sham- and sel-treated mice (**a**; 6 mice per group) and in sham- and L-DOPA-treated mice (**b**; 5 mice per group). Both drugs restore contextual fear memory in Tg2576 mice (**a**: two-tailed unpaired $t$-test: WT sham-Tg sham $*P = 0.015$, WT sel-Tg sham $*P = 0.025$, Tg sham-Tg sel $**P = 0.004$; (**b**): two-tailed unpaired $t$-test: WT sham-Tg sham $*P = 0.018$, WT L-DOPA-Tg sham $***P = 1.30 \times 10^{-4}$, Tg sham-Tg L-DOPA $**P = 0.007$). (**c**) Percentage of distance travelled in the target quadrant (previously containing the platform) during the Probe phase of the MWM test for 6-month-old sham- and sel-treated mice ($n = 5$ mice per group). Tg sham mice swam less in the target quadrant in comparison to the remaining groups, while selegiline was able to restore spatial memory performance (two-tailed unpaired $t$-test: WT sham-Tg sham $*P = 0.030$, WT sel-Tg sham $**P = 0.008$, Tg sham-Tg sel $*P = 0.031$). (**d**) Chocolate-induced place preference in 6-month-old sham- and sel-treated mice ($n = 4$ WT sham, 4 WT sel, 4 Tg sham and 6 Tg sel mice) showing mean time spent in paired and unpaired chambers in post-conditioning session, minus the time spent in the same chambers during the pre-conditioning session of a CPP test (two-way repeated measures ANOVA: chamber, $F_{1,14} = 36.97$, $P < 1.00 \times 10^{-4}$; chamber × treatment, $F_{3,14} = 231.34$, $P < 0.050$; treatment, $F_{3,14} = 0.50$, $P = 0.680$; WT sham-Tg sham $**P < 0.010$, WT sel-Tg sham $*P < 0.050$, Tg sham-Tg sel $***P < 0.001$ with Tukey's *post hoc* test). (**e**) Chocolate consumption during conditioning sessions of the CPP test shown in d (one-way ANOVA: $F_{3,14} = 37.10$, $P < 1.00 \times 10^{-4}$; WT sham-Tg sham $*P < 0.050$, WT sham-Tg sel $***P < 0.001$, WT sel-Tg sham $*P < 0.050$, WT sel-Tg sel $***P < 0.001$, Tg sham-Tg sel $***P < 0.001$ with Tukey's *post hoc* test). All data represent mean ± s.e.m.

the onset of VTA neuronal degeneration at 3 months of age. Accordingly, hippocampal synaptic plasticity and PSD composition, pyramidal neuron spine density, mnesic performances and reward processing, are all improved by stimulating the DA system with administration of L-DOPA or with reduction of endogenous DA degradation with selegiline. It is worth noting that, although we confirmed that selegiline treatment increased DA outflow in the hippocampus, we cannot exclude that the rescue of behavioural deficits might involve additional brain regions. Although the effectiveness of selegiline as a pharmacological agent in human patients is controversial[64], we used this inhibitor with the specific goal of enhancing the DAergic signal. It should be noted, however, that the effectiveness of selegiline could be dependent on the degree of VTA neurodegeneration.

Although the present observations are obtained in an experimental model of AD, they might provide an intriguing explanation to recent observations in AD patients[26] indicating that the clinical diagnosis of dementia is associated with early non-cognitive symptoms, such as depression and apathy. Thus, it is possible that early degeneration of VTA DAergic neurons could account for the worsening of memory as well as for at least part of the non-cognitive symptoms reported in early AD patients. In

fact, we provide evidence that the early neuronal apoptotic cell death observed in VTA DAergic neurons occurs at hippocampal pre-plaque stages in our animal model, and does not appear to result from local Aβ-plaque deposition in the VTA. Although we cannot exclude the accumulation of other Aβ aggregation forms in VTA neurons, we did not observe any abnormal pattern in the distribution of APPswe immunofluorescence that could suggest a link between intracellular Aβ and cell degeneration.

What is the mechanism underlying the selective vulnerability of VTA DAergic neurons to APPswe overexpression? The observation that DAergic neurons are lost in the VTA, while being spared in the SNpc, is a somehow symmetrical picture to what researchers involved in Parkinson's disease (PD) have been addressing for many years, trying to explain the selective loss of DAergic neurons in the SNpc, while neurons are largely spared in the VTA. A tempting hypothesis is that the selectivity of the midbrain cell population undergoing neuronal loss may be the result of a dying-back mechanism initiated at the axon terminals, whereby degenerative processes originating in the hippocampus and NAc cause VTA DAergic neuron loss in AD, while in PD analogous processes in the striatum lead to selective loss in the SNpc (ref. 65). Incidentally, a dying-back mechanism of

monoaminergic fibres without local amyloid pathology has been described in AD (refs 44,59,66–68). Alternatively, several lines of evidence suggest that mitochondrial dysfunction is present both in AD and PD. The diverse physiology of midbrain DAergic neurons might provide the answers needed to understand this mystery[69] making it possible that APPswe overexpression could have different effects in the two brain areas.

Although the molecular mechanisms underlying early VTA DAergic neuron degeneration remain to be elucidated, our results suggest the VTA as an important area of the brain to consider in the context of AD.

## Methods

**Animals and pharmacological treatments.** Heterozygous Tg2576 mice[21] and WT littermates were used at ages ranging from 2 to 11 months old, as described in the text. Only male subjects were used. Experiments were carried out in accordance with the ethical guidelines of the European Communities Council Directive (2010/63/EU). Experimental approval was obtained from the Italian Health Ministry (protocol #1191/2015PR).

Animals used for L-DOPA experiments were injected intraperitoneally (i.p.) with 10 mg kg$^{-1}$ L-DOPA plus 12 mg kg$^{-1}$ benserazide (both from Sigma-Aldrich) dissolved in saline, or with 0.9% saline only (sham) once a day for three consecutive evenings and 1 h before experimental session (Supplementary Fig. 4a). Animals used for selegiline experiments were subcutaneously injected with 3 mg kg$^{-1}$ selegiline (R-(−)-deprenyl hydrochloride, Sigma-Aldrich) prepared in 0.9% saline (sel) or with 0.9% saline only (sham). For a detailed description of the timeline of selegiline treatment for different experiments see Supplementary Figs 4c and 5a,d,g. For all treatments, injection volume was 5 μl per animal weight in grams.

**Immunohistochemistry and immunofluorescence.** Mice were anaesthetized with Rompun (20 mg ml$^{-1}$, 0.5 ml kg$^{-1}$, i.p., Bayer) and Zoletil (100 mg ml$^{-1}$, 0.5 ml kg$^{-1}$, Virbac) and perfused transcardially with 50 ml saline followed by 50 ml of 4% paraformaldehyde in phosphate buffer (PB; 0.1 M, pH 7.4). The brains were postfixed in paraformaldehyde overnight at 4 °C and then immersed in a 30% sucrose solution at 4 °C. The brains were cut into 30 μm-thick coronal sections using a freezing microtome.

The sections selected for immunohistochemistry (every second slice was processed for a total of 9 sections) were processed with a rabbit anti-TH antibody. The endogenous peroxidise was neutralized with a 0.3% $H_2O_2$ solution in PB. The sections were incubated overnight at 4 °C with the primary antibody diluted in PB containing 0.3% Triton X-100. After three washes in PB, sections were incubated with a biotinylated secondary antibody (Jackson Immunoresearch Laboratories, West Grove, PA, USA) followed by the avidin–biotin-peroxidase method (Vectastain, ABC kit, Vector, Burlingame, CA, USA) and using the chromogen 3,3′-diaminobenzidine (Sigma). Finally, sections were counterstained with Nissl staining, dehydrated and coverslipped with Entellan (Sigma).

For TH/GFAP, TH/Iba1, TH/NeuroTrace staining and for TH/APPswe, NeuroTrace/APPswe deposition, slices were incubated overnight with primary antibodies in PB containing 0.3% Triton X-100 and then incubated for 2 h at room temperature with secondary antibodies.

For DAT quantification, acute coronal brain slices (see below) were fixed in 4% paraformaldehyde in PB and transferred to 30% sucrose in PB at 4 °C. Slices were incubated for 2 days with primary antibodies in PB containing 1% Triton X-100 and then incubated for 2 h at room temperature with secondary antibodies. DAT levels (F/A) and TH levels (F/A) were quantified with ImageJ (http://imagej.nih.gov/ij/) as mean signal fluorescence intensity (F) on a defined area (A). Quantification was done on eight samples per mouse.

For the analysis of GFAP, Iba1 and DAT markers, images were taken as Z-stacks and these Z-stack images were then processed by maximum intensity projection. All samples were acquired with the same Z-stack thickness and the same laser settings. Data collection for densitometry was done by a researcher blind to the genotype of each animal.

Primary antibodies: TH (1:700, Millipore; MAB318; RRID: AB_2201528), GFAP (1:200, DAKO, Z0334; RRID: AB_2314535), hAPP695 (APPswe, 1:500, Biolegend, #803001; RRID: AB_2564653), DAT (1:200, Chemicon, MAB369; RRID: AB_2190413), Iba1 (1:400, Wako #019-19741; RRID: AB_839504). Secondary antibodies: Alexa Fluor 488 donkey anti-rabbit IgG (1:200; RRID: AB_2535792), Alexa Fluor 555 donkey anti-mouse IgG (1:200; RRID: AB_2536180) and Alexa Fluor 555 donkey anti-rabbit IgG (1:200; RRID: AB_2536182).

The sections were counterstained with NeuroTrace 640/660 deep-red Fluorescent Nissl Stain (1:200, Invitrogen), mounted using an anti-fade medium (Fluoromount, Sigma-Aldrich) and examined under a confocal laser-scanning microscope (LSM700, Zeiss). The specificity of the immunohistochemical labelling was confirmed by the omission of primary antibodies and the use of normal serum instead (negative controls).

All images were exported in tagged image file format (TIFF), contrast and brightness were adjusted and final plates were composed with Adobe Illustrator CS3.

**Stereological analysis.** Sections processed for immunohistochemistry were used for obtaining unbiased estimates of total number of TH$^+$ and TH$^−$ neurons in the SNpc and VTA and TH$^+$ neurons in the LC. The boundaries of these areas in the mouse brain were defined by TH staining, and area distinction was performed according to published guidelines[70]. We applied an optical fractionator stereological design (bilateral count) using the Stereo Investigator System (MicroBrightField Europe e.K.). A stack of MAC 5000 controller modules (Ludl Electronic Products, Ltd) was interfaced with an Olympus BX50 microscope with a motorized stage and a HV-C20 Hitachi digital camera with a Pentium II PC workstation. A three-dimensional optical fractionator counting probe (x, y, z dimension of 50 × 50 × 25 μm) was applied. The brain areas of VTA and SNpc were outlined using the 5 × objective and neuronal cells were marked with a ×100 oil-immersion objective. The brain area of LC was outlined using the ×5 objective and cells were marked with a ×40 oil-immersion objective. Data collection for cell counting was done by a researcher blind to the genotype of each animal.

Neurons were considered TH$^+$ if they showed cytoplasmatic TH immunoreactivity, while they were considered TH$^−$ if the nucleus was clearly visualized but cytoplasmic TH staining was absent.

The total neuronal number (TH$^+$ and TH$^−$ for VTA and SNpc; TH$^+$ for LC) was estimated according to the formula:

$$N = SQ \times 1/ssf \times 1/asf \times 1/tsf$$

where SQ represents the number of neurons counted in all optically sampled fields of the area of interest (VTA, SNpc or LC), ssf is the section sampling fraction, asf is the area sampling fraction and tsf is the thickness sampling fraction.

**In situ end labelling of DNA fragmentation (TUNEL).** For TUNEL staining, brains were removed as before and cut in coronal slices (10 μm thickness). Every tenth slice was processed for TUNEL. After removal of paraffin with xylene and rehydration in ethanol solutions of decreasing concentration, sections were digested with proteinase K (20 μg ml$^{-1}$, 15 min), washed in distilled water and exposed briefly to 3% $H_2O_2$. The TUNEL reaction was performed using the Apoptag Plus Peroxidase In-Situ Apoptosis Detection Kit (Millipore). TUNEL-positive cells were revealed with diaminobenzidine and $H_2O_2$ according to the supplier's instructions. Stained slices were lightly Nissl-counterstained with cresyl violet. Cells were defined as apoptotic if they were TUNEL-positive or if they showed typical nuclei with condensed chromatin or nuclear fragmentation or both. TUNEL-positive cells within the anatomical boundaries of VTA and SNpc (ref. 70) were counted at ×100 magnification with a Leitz DMRB microscope.

**Microdialysis.** Mice, anaesthetized with Zoletil and Rompun, were mounted on a stereotaxic frame (David Kopf Instruments) and implanted unilaterally with microdialysis probes 24–36 h before experiments. The concentric dialysis probes (AN69 fibres, Hospal Dasco) were implanted vertically at the level of the hippocampus (AP − 3.0, ML ± 2.7 from bregma)[70], NAc shell (AP + 1.6, ML ± 0.2) and striatum (AP + 1.0, ML ± 1.8). The probe lengths were 5 mm for hippocampus (3 mm membrane), 5.5 mm for NAc shell (1 mm) and 4.5 mm for striatum (2 mm). Each probe was fixed and the skin was sutured. Mice were returned to their home cages and the outlet and inlet probe tubing were protected by locally applied parafilm. Membranes were tested for in vitro recovery before surgery. On the day of the experiment each animal was placed in a circular cage containing microdialysis equipment: the microdialysis probe was connected to a CMA/100 pump (Carnegie Medicine) through PE-20 tubing and an ultra low torque multichannel power-assisted swivel (Model MCS5, Instech Laboratories) to allow free movement. Artificial cerebrospinal fluid (aCSF; in mM: NaCl 140; KCl 4; $CaCl_2$ 1.2; $MgCl_2$ 1) was pumped through the dialysis probe (2.1 μl min$^{-1}$). Following the start of the dialysis perfusion, mice were left undisturbed for ~1 h before the collection of three baseline samples to calculate the average basal concentration. Dialysate samples were collected every 20 min for 60 min. Brains were then postfixed in 4% paraformaldehyde, cut in coronal slices (100 μm) and processed for methylene blue staining. The correct positioning of the probes was confirmed under a microscope. Data from animals not showing proper placement were discarded.

Each dialysate sample (20 μl) was analysed by ultra-performance liquid chromatography. Concentrations (pg 20 μl$^{-1}$) were not corrected for probe recovery. The ultra-performance liquid chromatography apparatus (ACQUITY, Waters Corporation) was coupled to an amperometric detector (Decade II, Antec Leyden) containing an in situ Ag/AgCl reference electrode and an electrochemical flow-cell (VT-03, Antec Leyden) with a 0.7 mm glassy carbon electrode, mounted with a 25 mm spacer. The electrochemical flow-cell, set at a potential of 400 mV, was positioned immediately after a BEH C18 column (2.1 × 50 mm, 1.7 μm particle size; Waters Corporation). The column was kept at 37 °C (0.07 ml min$^{-1}$ flow rate). The composition of the mobile phase was (in mM): 50 phosphoric acid, 8 KCl, 0.1 EDTA, 2.5 1-octanesulfonic acid sodium salt, 12% MeOH and pH 6.0 adjusted with NaOH. Peak height obtained by oxidation of DA and noradrenaline was compared with that produced by a standard. The detection limit was 0.1 pg.

**Acute brain slice preparation for electrophysiology.** Acute brain slices (250–300 μm) were obtained following halothane anaesthesia and decapitation. The brain was rapidly removed and coronal slices containing the striatum and NAc core/shell or parasagittal slices containing the dorsal hippocampus were cut with a vibratome (VT1200S, Leica) in chilled bubbled (95% $O_2$, 5% $CO_2$) aCSF containing (in mM): NaCl 124, KCl 3, $NaH_2PO_4$ 1.25, $NaHCO_3$ 26, $MgCl_2$ 1, $CaCl_2$ 2, glucose 10 (~290 mOsm, pH 7.4). Slices were incubated for 1 h in aCSF at 32 °C and then transferred at room temperature for at least 30 min before recordings. A single brain slice was transferred to a recording chamber and completely submerged in aCSF (3–4 ml min$^{-1}$; 32 °C).

**Constant potential amperometry.** Amperometric detection of DA in acute brain slices containing the striatum and the NAc was performed using carbon fibre electrodes (diameter 30 μm, length 100 μm, World Precision Instruments) positioned near a bipolar Ni/Cr stimulating electrode, to a depth of 50–150 μm into the coronal slice. The imposed voltage (MicroC potentiostat, World Precision Instruments) between the carbon fibre electrode and the Ag/AgCl pellet was 0.55 V. For stimulation, a single rectangular electrical pulse was applied using a DS3 Stimulator (Digitimer) every 5 min along a range of stimulation intensities (20–1,000 μA, 20–40 μs duration). In response to a protocol of increasing stimulation, a plateau of DA release was reached at the maximal stimulation intensity (1,000 μA, 40 μs). Signals were digitized with Digidata 1440A coupled to a computer running pClamp 10 (both from Molecular Devices). Electrode calibration was performed at the end of each experiment by bath-perfused DA (0.3–10 μM).

**Multielectrode array recordings.** A parasagittal acute slice containing the dorsal hippocampus was placed over an 8 × 8 array of planar electrodes, each 50 × 50 μm in size, with an interpolar distance of 150 μm (MED-P5155, Alpha MED Sciences), adjusted so that the entire CA1 pyramidal layer and stratum radiatum were covered with underlying electrodes. The slice was kept submerged using a platinum ring covered with nylon mesh. Voltage signals were recorded with the MED64 System (Alpha MED Sciences) and digitized at 20 kHz followed by filtering at 0.1–1 Hz with a 6071E Data Acquisition Card (National Instruments), using Mobius software (Alpha MED Sciences). Field excitatory post-synaptic potentials (fEPSPs) were evoked by Schaffer collateral stimulation (100 μs duration) through one of the 64 planar electrodes placed in the stratum radiatum. The recording channel with the highest fEPSP amplitude, at a distance of at least 300 μm from the stimulation site, was chosen as recording site. Input–output curves were obtained by measuring the fEPSP initial slope at increasing 5 μA steps of afferent stimulation, delivered every 30 s. PPR was evaluated with pairs of stimuli, at half-maximal amplitude, separated by 20–500 ms. For LTP experiments, after at least 20 min of test stimulation (half-maximal shock, every 30 s) to assess fEPSP slope stability, the slice was challenged with a conditioning train at 100 Hz for 1 s, followed by test stimulation for at least 1 h. The degree of LTP was evaluated by the fEPSP mean slope at 55–60 min from the conditioning train, normalized to the mean slope during baseline. To study the acute effect of L-DOPA (Tocris) and of the selective DA receptor agonists SKF38393 (D1 receptor; Tocris) and quinpirole (D2 receptor; Tocris) on LTP, slices were continuously perfused throughout the experiment with aCSF containing the drugs at their final concentrations, prepared from concentrated stocks.

**Total protein extraction.** The striatum was dissected from acute coronal brain slices; the hippocampus was isolated from the entire brain. Tissues were homogenized in lysis buffer containing (in mM) 320 sucrose, 50 NaCl, 50 Tris-HCl pH 7.5, 1% Triton X-100, 1 sodium orthovanadate, 5 β-glycerophosphate, 5 NaF and protease inhibitor cocktail, incubated on ice for 30 min and centrifuged at 15,000 g for 10 min. The total protein content of the supernatant was determined by the Bradford method.

To evaluate APPswe levels, the hippocampus was dissected from parasagittal brain slices; NAc and striatum were dissected from coronal brain slices; VTA was dissected from horizontal brain slices. Tissues were homogenized in RIPA buffer containing (in mM) 50 Tris-HCl pH 7.5, 150 NaCl, 5 $MgCl_2$, 1 EDTA, 1% Triton X-100, 0.25% sodium deoxycholate, 0.1% SDS, 1 sodium orthovanadate, 5 β-glycerophosphate, 5 NaF and protease inhibitor cocktail, sonicated and incubated on ice for 20 min. The samples were centrifuged at 15,000 g for 20 min and the protein concentration of the supernatant was determined by the Bradford method.

**Place conditioning task.** To evaluate palatable food-induced CPP, we used a two-chamber apparatus made of two Plexiglas chambers (15 × 15 × 20 cm) that differ in the pattern and the colour of the grid, connected by a central compartment (15 × 5 × 20 cm) with two sliding doors (4 × 20 cm). In each chamber two triangular parallelepipeds (5 × 5 × 20 cm), made of black Plexiglas and arranged in various patterns, were used as conditioned stimuli. The lighting conditions (visual and tactile cues) were adjusted to prevent preference for a certain chamber. On the pre-conditioning day, each mouse was left in the central alley and allowed free access and exploration of the adjacent chambers of the apparatus, in the absence of food for 15 min. During the following 6 days (conditioning session), each mouse

was confined daily for 30 min alternately in one of the two chambers. One of the patterns was consistently paired with palatable food (0.5 g milk chocolate, Milka Alpine Milk Chocolate providing 5.31 kcal g$^{-1}$ of energy; paired chamber) and the other with regular chow (RC, Mucedola 4RF21 diet; unpaired chamber). The RC pellet and the chocolate portion provided each day were designed to be isocaloric. Animals were randomly assigned to consume either chocolate or control diet. For each genotype group pairings were counterbalanced so that in half of the group chocolate was paired with one of the patterns and in the other half with the other. Testing was conducted on post-conditioning day using the pre-conditioning procedure during which the total time spent in each chamber was recorded. Animal behaviour was recorded with a CCD video camera, the signal was digitized and transferred to a computer. Data were analysed with the EthoVision XT software (Noldus) to obtain the time spent during pre- and post-conditioning sessions for each subject, used as raw data for preference scores in each sector of the apparatus. Chocolate consumption was assessed by averaging, after weighting, the residual amount of food at the end of chocolate-receiving days.

**Contextual fear-conditioning test.** The apparatus consisted of a 21 × 21 × 49 cm chamber (Ugo Basile) with grey Plexiglas walls and transparent ceiling to allow video recordings. The grid floor (steel pieces spaced by 1.5 cm) was connected to a shock generator scrambler. The CFC test encompassed two sessions: training and context test. During training each mouse was placed in the chamber for 2 min of free exploration, followed by three non-signalled foot shocks (1 mA, 2 s duration, 60 s intervals) delivered through the grid floor. After 24 h, the subject was placed again for 5 min in the training chamber receiving no foot shock (context test). During training and context test the mouse behaviour was recorded. To evaluate aversive learning and memory, the total duration of freezing (behavioural immobility, except for respiration movements) exhibited during training and context test was manually scored by an observer blind to the experimental group of the specimen, through EthoVision XT software (Noldus).

**PSD preparation.** For sub-synaptic fractionation of PSD, hippocampi were homogenized with a Dounce tissue grinder in homogenization buffer containing (in mM) 320 sucrose, 10 Tris-HCl pH 7.4, 1 EDTA, 1 $NaHCO_3$, 1 PMSF, 1 sodium orthovanadate, 5 NaF, 20 β-glycerophosphate and protease inhibitor cocktail. The homogenate was centrifuged at 1,000 g (10 min) and the supernatant was again centrifuged at 10,000 g (15 min). The pellet was homogenized in homogenization buffer containing 0.5% Triton X-100 with a Dounce tissue grinder, incubated 40 min on ice and centrifuged at 32,000 g (20 min). The resulting pellet containing PSDs was processed for protein extraction. Briefly, the pellet was resuspended in RIPA buffer containing (in mM) 50 Tris-HCl pH 7.5, 150 NaCl, 5 $MgCl_2$, 1 EDTA, 1% Triton X-100, 0.25% sodium deoxycholate, 0.1% SDS, 1 sodium orthovanadate, 5 β-glycerophosphate, 5 NaF and protease inhibitor cocktail, was sonicated and incubated on ice for 20 min. The samples were centrifuged at 11,500 g for 10 min and the protein concentration of the supernatant was determined by the Bradford method.

**Immunoblotting analysis.** Proteins were applied to SDS–PAGE and electroblotted on a polyvinylidene difluoride membrane. Immunoblotting analysis was performed using a chemiluminescence detection kit. The relative levels of immunoreactivity were determined by densitometry using the ImageJ software. Primary antibodies: GluA1 (1:1,000; Millipore, 04-855), GluA1p-Ser845 (1:1,000; Millipore, AB5849; RRID: AB_92079), PSD-95 (1:1,000; Millipore, MAB1598; RRID: AB_94278), GluN2A (1:250; Santa Cruz, sc-1468; RRID: AB_670223), Actin (1:25,000; Sigma-Aldrich, A5060; RRID: AB_476738), D1 (1:1,000, Abcam, ab20066; RRID: AB_445306), D2 (1:1,000, Millipore, AB5084P; RRID: AB_2094980), αCaMKII (1:1,000, ThermoFisher, #13-7300; RRID: AB_2533032), TH (1:1,000, Abcam, Ab112; RRID: AB_297840), hAPP695 (APPswe, 1:500; Biolegend, #803001; RRID: AB_2564653). Secondary antibodies: goat anti-mouse IgG (1:3,000; Bio-Rad; RRID: AB_11125936), goat anti-rabbit IgG (1:3,000; Bio-Rad; RRID: AB_11125142) and rabbit anti-goat IgG (1:3,000; Bio-Rad; RRID: AB_11125144).

Membranes were stripped using Re-Blot Plus Strong Solution (Millipore) for 15 min at room temperature. Full blots are shown in Supplementary Information.

**Dendritic spine density analysis.** To study the morphology of dendritic spines in hippocampal pyramidal neurons, brains were removed from the skull and impregnated with a Golgi-Cox solution. Brains were first immersed in a 5% $Cr_2K_2O_7$, 5%$Cl_2Hg$ and 5% $CrK_2O_4$ solution (Sigma-Aldrich) for 6 days, then moved to a 30% sucrose solution for 3–5 days and finally sliced in 100 μm coronal sections at the CA1 level (−2.3 to −5.8 mm from bregma). Slices were sequentially washed with the following solutions: distilled water (1 min), $H_5NO$ (Sigma-Aldrich; 30 min in the dark), distilled water (1 min), Kodak Fix-film (Sigma-Aldrich; 30 min in the dark), distilled water (1 min), sequentially in 50, 70 and 95% alcohol (1 min each), twice in 100% alcohol (5 min), in a solution of one-third xylene, one-third chloroform and one-third 100% alcohol (15 min), xylene (15 min). Slices were then coverslipped with Canada Balsam. The stained slices were analysed under a × 100 oil-immersion objective (Axioskop, Zeiss). During morphological analysis (Neurolucida v11, MicroBrightField) the specimen identity was unknown. A hippocampal pyramidal neuron was processed for

morphological analysis if labelling was uniform, reaction precipitates were absent, if it did not overlap with neighbouring cells, if spines were clearly visible, dendritic arbours were relatively parallel to the section plane and arbours in distal dendritic branches were intact and visible. With these criteria, 30 CA1 pyramidal neurons were selected per group (WT sham, Tg2576 sham, WT sel and Tg2576 sel). The basal and apical dendritic arbours of each cell were examined separately using the software's Sholl Analysis tool. Terminal spine density was calculated as number of spines (defined as protrusions of the dendritic membrane regardless of shape) along a 25 μm dendritic terminal segment.

**Morris water maze.** Mice were placed in a circular white pool (diameter 145 cm) filled with 23 ± 2 °C water made opaque by the addition of non-toxic acrylic white colour (Giotto). The protocol consisted of an 8-trial Cue phase (1 day) and a 20-trial Place phase followed by a 1-trial Probe phase (4 days). During Cue phase, a visible platform (5 cm diameter) emerged 0.5 cm over the water level and surmounted by small red plastic caps (to make it more evident) was placed in the middle of the pool. During Place phase, the escape platform (5 cm diameter) submerged 0.5 cm under the water level was placed in the middle of the north–west quadrant 20 cm from the side walls. The pool was positioned in a uniformly lighted room with various external cues and surmounted by a camera that relayed signals to a monitor and to the image analyzer (EthoVision XT, Noldus). The Cue phase was composed of two consecutive sessions (four 60 s-trials each; inter-trial interval: 20 min; and inter-session interval: 2 h). The Place phase was composed of four daily sessions (five 60 s-trials per day; inter-trial interval: 20 min). During the inter-trial and inter-session intervals, mice were placed in their home cages. At the start of each trial, mice were put into the pool at different quadrants for each subject and then retrieved after finding the visible or hidden platform. The mice that failed to find the platform within 60 s, were helped by the experimenter. When mice climbed the platform, they were allowed to remain on it for 20 s. To evaluate spatial memory, 1 h after the last trial of Place phase mice were subjected to the Probe phase, in which the platform was removed and the mice (released from the opposite side of the pool) were allowed to search for it for 30 s. Data analysis were performed using the following parameters: time spent to reach the platform during Cue and Place phases; percentage of distance swum in the target quadrant (previously containing the platform) during Probe phase.

**Statistical analysis.** CPP: data were analysed by two-way analysis of variance (ANOVAs) with the genotype (Tg2576 versus WT) and chambers (paired versus unpaired) as independent factors. CPP data for selegiline treatment were analysed by two-way ANOVAs for treatment (sham versus sel) and chambers (paired versus unpaired). Chocolate consumption after selegiline was analysed with one-way ANOVA. *Post hoc* comparisons were performed with STATISTICA software (v8.0, StatSoft, Inc., 2007) using Tukey's HSD test.

Microdialysis: data were analysed with one-way repeated measures ANOVAs with genotype (Tg2576 versus WT) as between-subject factor and time (20 min blocks, 3 levels) as within-subject factor. Microdialysis data in Tg2576 mice following selegiline treatment were analysed with one-way repeated measures ANOVAs for treatment (Tg sham versus Tg sel) and time (20 min blocks, 3 levels).

LTP: data for sub-chronic L-DOPA and selegiline treatment were analysed by two-way ANOVAs for genotype (Tg2576 versus WT) and treatment (sham versus drug). LTP data for the effect of bath administration of L-DOPA and DA receptor agonists on Tg2576 mice were analysed with one-way ANOVA. In both cases *post hoc* comparisons were assessed with Bonferroni's test.

All other data were analysed with a two-tailed paired or unpaired Student's *t*-test, as described in Figure legends. Data are presented as mean ± s.e.m. Except for CPP, all statistical analysis was performed using GraphPad Prism (v5.00). Values of $P \leq 0.05$ were considered to be statistically significant (shown in figures as $*P \leq 0.05$, $**P \leq 0.01$ and $***P \leq 0.001$).

**Sample size.** The number of samples in each group for immunohistochemical, immunofluorescent, stereological, microdialysis, electrophysiological, biochemical, behavioural and morphological evaluations were determined based on published studies. The experimenter performing surgeries was known to hit the targets used (NAc shell, striatum or hippocampus).

**Randomization.** All randomization was performed by an experimenter. Animals used in all experiments were selected randomly. For CPP, animals were also counterbalanced across groups.

**Data availability.** The data that support the findings of this study are available from the corresponding author (Marcello D'Amelio; m.damelio@unicampus.it) upon reasonable request.

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

## Acknowledgements

M.D.A. was supported by the Italian Ministry of Health (Progetto Giovani Ricercatori Project Code GR-2011-02351457) and by the Alzheimer's Association (NIRG-11-204588). We are also grateful to Associazione C.M.N. Sport (ONLUS) for the fellowship offered to P.K. We thank Drs Riviello and Wirz for assistance with animal caring.

## Author contributions

M.D.A. conceived and designed the study, supervised all of the experiments. A.N., V.C. and M.C.D.A. designed and carried out the molecular biology experiments. M.T.V., A.N. and R.M. performed cell counting and immunohistochemical analysis. F.R.R. and M.F. performed and analysed amperometric experiments. D.C., P.D.B. and L.P. performed and analysed CFC and MWM experiments and dendritic spine counts. S.P.A., E.C.L. and M.S. performed and analysed microdialysis experiments. G.G. performed and R.C. designed, analysed and interpreted CPP experiments. N.B., D.A., A.C. and P.K. performed field recordings. N.B. and N.B.M. supervised the electrophysiology experiments. M.D.A, F.K., P.K., R.C., N.B. and A.N. wrote the manuscript. All authors discussed results and commented on the manuscript.

## Additional information

**Competing interests:** The authors declare no competing financial interests.

