## [Peer Review File · Nature Communications]

Reviewers' comments:

Reviewer #1 (Remarks to the Author):

This is a very relevant paper that together with recent published papers highlights the importance of new dopamine-based treatment strategies for Alzheimer's disease. This work describes a dysfunction in the dopaminergic neurotransmission in the Tg2576 transgenic model for Alzheimer's disease. The authors described a progressive death of TH+ neurons in the VTA that correlates with aging in the Tg2576 model, while TH+ neurons in the SN remained unaffected. The loss of TH+ neurons in the VTA resulted in decreased basal levels of dopamine in the hippocampus and in the NAc shell. These DA reduced levels were correlated to deficiencies in hippocampal LTP maintenance, hippocampal-related memory performance and impaired food reward processing.

Although in general, the paper is well written and with straightforward conclusions, some of the main findings are not new. In addition, there are some major issues that should be addressed or discussed to improve this paper.

There are several directly related papers that are not cited nor discussed in this work. In 2008, Liu Y. and collaborators described selective neurodegeneration of TH+ neurons in the VTA but not in the SN in the APP^{swE}/PS1 E9 mouse model, which was correlated with anxiety-associated behavior. They found that TH+ neurons loss is due to a progressive degeneration process that starts with axonal degeneration related to amyloid beta accumulation in cortical and hippocampal areas that are innervated by the VTA. Conversely, TH+ neurons in the SN remained unaffected, since DA projections from the SN does not seem to be affected by amyloid beta. This is supported by the work of Moreno-Castilla and coworkers, where they showed that VTA cortical terminals are damaged by AB oligomers. These references may explain the selective VTA degeneration that authors in the present work consider as "still to be explored", and they should be discussed. Therefore, it would be important to evaluate the integrity of TH+ terminals and the accumulation of AB oligomers or plaques in the hippocampus, nucleus Accumbens shell and core, and in the striatum.

Similarly, there are several related papers that should be further discussed, like the work of Martorana and Koch (2005), and the studies showing that activation of dopamine receptors restore LTP in the hippocampus (see work of Jurgensen et. al., 2011) and rescues memory deficiencies in hippocampal- (Himeno et. al., 2011) and cortical- (Guzman-Ramos et. al., 2012) related memories.

The authors propose that cell loss in the VTA does not appear to result from extracellular AB deposition, as plaque deposition seems to be scarce in the VTA (Figure 2c). However, the authors should determine whether any other aggregation form of amyloid beta (like oligomers) is present in both VTA and SN.

Authors performed two-tailed t-test in order to evaluate differences between LTP in WT and Tg animals, and between treatments. However, a repeated measures ANOVA would be a better tool to evaluate these effects and for group comparisons (Figure 5a and 5b).

Finally, the authors report that selegiline treatment restores the performance of Tg mice in CFC task and rescues hippocampal PSD composition, is it the same for levodopa treatment?

Reviewer #2 (Remarks to the Author):

In this manuscript, Nobili and colleagues describe age-dependent loss of dopaminergic neurons in the VTA, but not in the SNpc, in Tg2576 mice, which overexpress a mutant form of human amyloid

precursor protein (APP695swe). The authors show that this neuronal loss starts at pre-plaque stages of the disease, and results in reduced dopamine outflow in the hippocampus and in the NAc shell. Accordingly, with the progressive dopaminergic cell death, the Tg2576 mice exhibit decreased hippocampal synaptic function and memory with aging, as well as impairments in food reward processing at 6 months of age. In addition, biochemical analysis of PSD fractions reveals significant alterations in the expression of postsynaptic D1 dopamine receptors in the hippocampus. The authors then utilize L-DOPA (a precursor of dopamine) and Selegiline (MAO inhibitor) to restore the dopaminergic drive in the dorsal hippocampus of 6 month-old mice. The treatment with either L-DOPA or Selegiline rescues deficits in hippocampal synaptic plasticity (LTP) and impairments in contextual fear conditioning in Tg2576 mice, possibly due to a restored dopaminergic drive in the hippocampus. Furthermore, Selegiline administration leads to a restoration of hippocampal PSD composition, in particular the phosphorylation levels of GluA1. Finally, administration of Selegiline also rescues abnormal spine density in pyramidal neuron apical dendrites located in stratum radiatum of area CA1.

Although the Tg2576 mice represent a well-established mouse model of AD, this study presents an interesting way to better understand the role played by the dopaminergic neuron population of the VTA in AD-related memory dysfunction. In addition, the authors provide useful evidence of an early and selective degeneration of VTA dopaminergic neurons that is correlated with onset of impaired hippocampal function. Nevertheless, there are several important issues that remain to be addressed.

Major Comments

- 1) The authors provide no biochemical evidence that the rescue of hippocampal LTP in Tg2576 mice due to administration of either L-DOPA or Selegiline is because of increased dopamine availability in the hippocampus. Both the treatments act by increasing the endogenous levels of dopamine, but this does not necessarily mean that there is increased dopaminergic release in the hippocampus. The standard approach for addressing this issue is to use *in vivo* microdialysis to detect changes in extracellular dopamine in hippocampus also after the aforementioned treatments.
- 2) In light of the inconsistent results of Selegiline in clinical treatment and the dependence of its efficacy on the remaining dopaminergic neurons in the VTA, the authors' motivation in utilizing this compound as a modulator of dopamine activity is unclear. It is well-known that direct stimulation of dopamine D1/D5 receptors by using dopaminergic agonists rescue hippocampal LTP impairment in different models of AD and may represent a valid alternative.
- 3) Although an increase in the levels of D1 receptors in the PSD fractions from the hippocampus can be explained as a result of reduced dopamine availability at the postsynaptic sites, it may also be that at glutamatergic synapses the expression of D1 receptors is regulated by either direct or indirect interactions with different postsynaptic partners such as NMDARs and AMPARs, as well as postsynaptic scaffolding proteins such as PSD95. Therefore, a better measure of both the health of the VTA dopaminergic neurons and dopamine availability would be to determine the expression of D2 auto-receptors located at presynaptic sites, as well as in the expression of DAT in hippocampus, especially since no changes in D2 postsynaptic expression was determined.
- 4) By using amperometric recording the authors observe no difference in evoked dopamine in dorsal striatum. Curiously, the authors did not examine any motor behavior(s) in these mice despite reports that multiple AD mouse models show altered transient motor performance that could be attributed to dopaminergic dysfunction on the nigro-striatal pathways. *In vivo* microdialysis would be more helpful in detecting basal levels of dopamine in that area as well as in the NAc core.
- 5) A minor point – the symbols for statistical significance are inconsistent in the figures and figure

legends.

Reviewer #3 (Remarks to the Author):

The authors of this paper provide evidence that in a particular transgenic Alzheimer disease model there is a substantial loss of ventral tegmental area (VTA) dopamine (DA) neurons prior to the onset of plaque formation elsewhere in the brain. They go on to provide some supporting data indicating that some of reward-related and cognitive defects may be due to loss of DA signaling in the nucleus accumbens (NAc) and hippocampus. Their rescue of some of the phenotypes with L-DOPA and selegiline is remarkable. While intriguing, there are a number of holes in this story that need to be addressed.

1. The loss of TH-expressing neurons in the VTA appears to be well substantiated, although peculiar, because substantia nigra (SN) neurons are generally thought to be more vulnerable to cell death. The authors recognize this issue, but provide no plausible explanation as to why VTA neurons might be more vulnerable to the transgene in this model. In fact, they don't even show whether the transgene is expressed in the midbrain or selectively in VTA neurons.
2. The authors look at astrocyte activation in the VTA by staining for GFAP, but they should also measure microglia activation with Iba1 staining.
3. The authors show that there is less evoked DA release in the NAc shell but not NAc core, which is peculiar since both regions of the NAc are innervated preferentially by VTA neurons. The authors account for this difference by saying that the core receives input from the SN and provide 3 references (28-30). Two of the references are reviews that do not make this distinction and the third talks about role of shell and core but does not address the source of DA in those two regions. The authors need to provide a more compelling argument (or data) showing that DA in NAc core comes from the SN. There may well be sub-sets of VTA DA neurons, some of which project to core and some to shell, and those that project to the shell might be more vulnerable to transgene expression.
4. The authors claim that they cannot measure evoked DA release in the hippocampus and resort to microdialysis. However, they measure just as much DA by microdialysis in hippocampus (Fig. 4a) as they do in NAc (Fig. 3d); thus, they should also be able to measure evoked DA release. Most of the TH in the hippocampus comes from the locus ceruleus (LC), so it is not possible to conclude that the deficiency in hippocampal TH is due to loss of VTA neurons (Fig. 4c). There is a fairly persuasive paper indicating that most of the DA in the hippocampus actually comes from LC neurons (Smith, *J Neurosci* 32, 6072) so authors need to take that into account in their analysis and discussion.
5. There seems to be some discrepancy in the analysis of hippocampal LTP which they say is normal at 2 and 3 months of age, even though there is already significant loss of VTA DA neurons at 3 months (Fig. 1).
6. The behavioral analysis is rather sparse. If the authors really want to examine reward learning they need to do more than a conditioned place preference experiment. Pavlovian and instrumental learning experiments along with progressive ratio experiments should be included. And, the authors need to examine whether the drug treatments restore normal behavior in their transgenic model.
7. Likewise, there are additional hippocampal dependent tasks should be included, e.g. Morris water maze. The contextual fear conditioning (CFC) data (Fig. 4f, 6e) are meaningless since the authors show that the control mice spent only 20 seconds 'freezing' after conditioning during a 5-min test period (p. 29 line 7). Most authors obtain 70-80% of time spent freezing with this paradigm.

POINT-BY-POINT REPLY TO REVIEWER #1

After a short statement on the importance of our work, the Reviewer raised the following critiques:

Point 1-2 raised by Reviewer #1:

There are several directly related papers that are not cited nor discussed in this work.

In 2008, Liu Y. and collaborators described selective neurodegeneration of TH⁺ neurons in the VTA but not in the SN in the APP^{swe}/PS1 E9 mouse model, which was correlated with anxiety-associated behavior. They found that TH⁺ neurons loss is due to a progressive degeneration process that starts with axonal degeneration related to amyloid beta accumulation in cortical and hippocampal areas that are innervated by the VTA. Conversely, TH⁺ neurons in the SN remained unaffected, since DA projections from the SN do not seem to be affected by amyloid beta. This is supported by the work of Moreno-Castilla and coworkers, where they showed that VTA cortical terminals are damaged by AB oligomers. These references may explain the selective VTA degeneration that authors in the present work consider as "still to be explored", and they should be discussed.

Therefore, it would be important to evaluate the integrity of TH⁺ terminals and the accumulation of AB oligomers or plaques in the hippocampus, nucleus Accumbens shell and core, and in the striatum.

Similarly, there are several related papers that should be further discussed, like the work of Martorana and Koch (2005), and the studies showing that activation of dopamine receptors restore LTP in the hippocampus (see work of Jurgensen et. al., 2011) and rescues memory deficiencies in hippocampal- (Himeno et. al., 2011) and cortical- (Guzman-Ramos et. al., 2012) related memories.

Reply:

Following the Reviewer's recommendations we revised the bibliography and the Discussion section to include the missing references. We believe that these references strengthen our results of selective VTA neurodegeneration, in particular, the work by Liu et al., (J. Neurosci., 2008) in which the Authors showed that the progressive axonal degeneration is correlated with A β accumulation in the projection areas, and the work of Moreno-Castilla et al., (Neurobiology of Aging, 2016) showing that injection of A β oligomers in the insular cortex of WT animals is sufficient to damage

VTA cortical terminals. The hypothesis that the selectivity in VTA DAergic degeneration in our model likely arises from a dying-back mechanism starting from the axonal terminals in projection areas is now discussed further in the Discussion.

The novelty of our work in respect to these works is the observation that the selective loss of VTA DAergic neurons, starting at 3-months in Tg mice, in absence of TH⁺ neuron loss from the SNpc and the Locus coeruleus (see new data in Fig. 4c,d), occurs at hippocampal pre-plaque stages. This is now clearly shown in new Supplementary Fig. 1 where, driven by the Reviewer's request, we evaluated the accumulation of extracellular A β plaques, intracellular APP_{swe} staining and total levels of APP_{swe} protein in the hippocampus, VTA, SNpc, NAc (core and shell) and striatum in 6-month-old mice (new Supplementary Fig. 1a,b), in comparison with A β plaque deposition in the hippocampus of aging (11-month-old) mice (Supplementary Fig. 1c).

Moreover, as requested by the Reviewer, we evaluated the integrity of TH⁺ terminals in the hippocampus. The new Fig. 4f clearly demonstrates a strong reduction of TH staining in the transgenic hippocampus, in strict agreement with the reduction of hippocampal TH protein expression detected by western blot analysis (Fig. 4e).

The recommended papers and other research works pointing out the relationship between DA-related pharmacological manipulations and improved memory function are now reported in the reference list and discussed in the new version of the manuscript.

Point 3 raised by Reviewer #1:

The authors propose that cell loss in the VTA does not appear to result from extracellular A β deposition, as plaque deposition seems to be scarce in the VTA (Figure 2c). However, the authors should determine whether any other aggregation form of amyloid beta (like oligomers) is present in both VTA and SN.

Reply:

As depicted in new Supplementary Fig. 1a that examines the accumulation of A β , we did not detect focal accumulation of staining in the cytoplasm of VTA or SNpc DAergic cells. Instead, intracellular staining was rather diffuse and regular in these neurons, providing evidence for the absence of intracellular aggregation. This point is now further discussed in the text.

The diffuse staining in the VTA appears to be different from the more intense and focal staining in the hippocampus (Supplementary Fig. 1a) and APP_{swe} levels are significantly lower in the VTA (Supplementary Fig. 1b) in line with the hypothesis of a dying-back mechanism starting from the axonal terminal in projections areas.

Point 4 raised by Reviewer #1:

Authors performed two-tailed t-test in order to evaluate differences between LTP in WT and Tg animals, and between treatments. However, a repeated measures ANOVA would be a better tool to evaluate these effects and for group comparisons (Figure 5a and 5b).

Reply:

Following the Reviewer's request, LTP data in WT and Tg mice are now analysed with two-way ANOVAs (with genotype and treatment as independent factors) followed by Bonferroni's post-hoc tests. The Figure legend and statistical analysis section in Methods are now changed accordingly.

Point 5 raised by Reviewer #1:

Finally, the authors report that selegiline treatment restores the performance of Tg mice in CFC task and rescues hippocampal PSD composition, is it the same for levodopa treatment?

Reply:

Driven by the Reviewer's request we studied the effect of L-Dopa treatment in the CFC task and in the recovery of the composition of hippocampal post-synaptic densities (PSD) in 6-month-old animals. We found (**new data reported in Fig. 7b**) that L-Dopa treatment is able to restore the performance of Tg mice in CFC, in agreement with an earlier report¹. Moreover, we show that the same treatment is able to rescue hippocampal PSD composition (**new Fig. 6c**). These new data, together with a rescue of spatial memory deficits by selegiline in Tg mice during a Morris water maze test (**new Fig. 7c**), confirm that the enhancement of the DAergic tone can improve memory function in Tg2576 mice.

1. Ambrée, O. *et al.* Levodopa ameliorates learning and memory deficits in a murine model of Alzheimer's disease. *Neurobiol. Aging*. **30**, 1192-204 (2009).

POINT-BY-POINT REPLY TO REVIEWER #2

After a short introduction on the importance of our work, the Reviewer raised the following comments:

Point 1 raised by Reviewer #2:

The authors provide no biochemical evidence that the rescue of hippocampal LTP in Tg2576 mice due to administration of either L-DOPA or Selegiline is because of increased dopamine availability

in the hippocampus. Both the treatments act by increasing the endogenous levels of dopamine, but this does not necessarily mean that there is increased dopaminergic release in the hippocampus. The standard approach for addressing this issue is to use in vivo microdialysis to detect changes in extracellular dopamine in hippocampus also after the aforementioned treatments.

Reply:

In response to the Reviewer's request and according to this suggestion we used *in-vivo* microdialysis to measure DA basal outflow in the hippocampus of Tg2576 mice following selegiline treatment. We show that selegiline treatment is indeed able to increase the basal outflow of DA in the hippocampus. These new data have now been included in **Supplementary Fig. 4d**.

Point 2 raised by Reviewer #2:

In light of the inconsistent results of Selegiline in clinical treatment and the dependence of its efficacy on the remaining dopaminergic neurons in the VTA, the authors' motivation in utilizing this compound as a modulator of dopamine activity is unclear. It is well-known that direct stimulation of dopamine D1/D5 receptors by using dopaminergic agonists rescue hippocampal LTP impairment in different models of AD and may represent a valid alternative.

Reply:

We think the Reviewer is correct about his/her concerns on selegiline, and we share these considerations (this is acknowledged in the Discussion where we also discuss that our rationale for selegiline was to use it merely as a tool for enhancing the DAergic signal). That was why, additionally to selegiline, we also tested the effects of sub-chronic treatment with _L-DOPA as an additional tool of study.

In line with the Reviewer's suggestion, we performed new experiments on Tg mice using acute administration of SKF38393 to selectively activate D1/D5 receptors and these results were comparable with those of acute _L-DOPA administration in rescuing hippocampal LTP impairment. Instead, quinpirole acting on D2 receptors had no effect on LTP. These data are now in **new Supplementary Fig. 4b**. The text and Methods sections are changed accordingly.

Point 3 raised by Reviewer #2:

Although an increase in the levels of D1 receptor in the PSD fractions from the hippocampus can be explained as a result of reduced dopamine availability at the postsynaptic sites, it may also be that at glutamatergic synapses the expression of D1 receptors is regulated by either direct or indirect interactions with different postsynaptic partners such as NMDARs and AMPARs, as well as postsynaptic scaffolding proteins such as PSD95. Therefore, a better measure of both the health of the VTA dopaminergic neurons and dopamine availability would be to determine the expression of D2 auto-receptors located at presynaptic sites, as well as in the expression of DAT in hippocampus, especially since no changes in D2 postsynaptic expression was determined.

Reply:

The evidence that the Tg hippocampus is DA-denervated is reported in the Fig. 4: the strong reduction of TH staining (**new data in Fig. 4f**) together with the reduced expression of TH protein (Fig. 4e) is a measure of reduced availability of VTA DAergic projections. These results, together with the microdialysis data showing a reduced DA availability (Fig. 4a), strongly indicate that the increased level of D1 receptors is a compensatory result due to reduced DA availability. Furthermore, we now show that the selegiline treatment leads to an increase of basal DA outflow in the Tg2576 hippocampus (**new data in Supplementary Fig. 4d**) and in response to this we found a reduced expression of D1 receptor, as a compensative response (Fig. 6b).

Finally, the evidence that quinpirole displayed no effect on LTP (**new data in Supplementary Fig. 4b**) might suggest no changes in the expression of D2 auto-receptors located at presynaptic sites.

Point 4 raised by Reviewer #2:

By using amperometric recording the authors observe no difference in evoked dopamine in dorsal striatum. Curiously, the authors did not examine any motor behavior(s) in these mice despite reports that multiple AD mouse models show altered transient motor performance that could be attributed to dopaminergic dysfunction on the nigro-striatal pathways. In vivo microdialysis would be more helpful in detecting basal levels of dopamine in that area as well as in the NAc core.

Reply:

According to the Reviewer's suggestion, we measured by means of *in vivo* microdialysis the basal levels of DA in the striatum (new Fig. 3e) and we found a significant reduction of basal DA. Our data show that: (i) the number of SNpc DAergic neurons is unaltered in Tg mice (Fig. 1b), (ii) in this brain region we did not detect increased cell apoptosis, astrogliosis or microgliosis (Fig. 2a-c), (iii) TH protein levels, as a measure of DAergic innervation in the striatum, are unchanged (Fig. 4g) and (iv) the evoked DA release in the striatum is also unchanged (Fig. 3b). Together, these data support the idea that the nigrostriatal pathway in Tg mice is intact, at least at the age used in the study (3-6 month-old mice).

Rather, we attribute the reduced basal striatal levels of DA to alterations in the mechanisms controlling tonic DA release. VTA DAergic neurons project both to the prefrontal and primary motor cortex¹⁻³ where they regulate corticostriatal glutamate release onto striatal cholinergic interneurons that, in turn, mediate the tonic release of DA from nigrostriatal terminals^{4,5}. We believe that in the Tg brain, where VTA DAergic neurons degenerate early, DAergic inputs to the cortex - and consequently, the cortical control of striatal DA release - are likely to be affected, hence explaining the reduction we observed in tonic DA levels. In fact, a situation of normal phasic nigrostriatal DA release (favoring low-affinity D1 receptor activation) and reduced corticostriatal-

regulated tonic DA release (which would reduce/inhibit the activation of high-affinity D2 receptors), would favor the hyperactivation of the direct pathway without the fine-control of the indirect pathway on movement^{6,7} and, together with deficits in the prefrontal and primary motor cortex, could explain the altered motor behavior usually observed in Tg mice⁸.

Of note, unlike what is routinely performed in the rat, the small size of the mouse NAc core does not permit reliable measurements of DA by means of microdialysis as it is challenging to place probes exclusively in the core subregion in the mouse model; for this reason we have chosen to perform only amperometric recordings on brain slices containing the NAc core.

1. Karreman, M. & Moghaddam, B., The prefrontal cortex regulates the basal release of dopamine in the limbic striatum: an effect mediated by ventral tegmental area. *J. Neurochem.* **66**, 589–598 (1996).
2. Hosp, J.A. *et al.* Dopaminergic projections from midbrain to primary motor cortex mediate motor skill learning. *J. Neurosci.* **31**, 2481–2487 (2011).
3. Kunori, N. *et al.* Voltage-sensitive dye imaging of primary motor cortex activity produced by ventral tegmental area stimulation. *J. Neurosci.* **34**, 8894–8903 (2014).
4. Rice, M.E. *et al.* Dopamine release in the basal ganglia. *Neuroscience* **198**, 112–137 (2011).
5. Kosillo, P. *et al.* Cortical Control of Striatal Dopamine Transmission via Striatal Cholinergic Interneurons. *Cereb. Cortex* 2016 doi: 10.1093/cercor/bhw252
6. Gerfen, C.R. & Surmeir, D.J. Modulation of striatal projection systems by dopamine. *Annu. Rev. Neurosci.* **34**, 441–466 (2011).
7. Onn, S.P. *et al.* Dopamine-mediated regulation of striatal neuronal and network interactions. *Trends Neurosci.* **23**, (10 Suppl):S48–56 (2000).
8. Webster, S.J. *et al.* Using mice to model Alzheimer's dementia: an overview of the clinical disease and the preclinical behavioral changes in 10 mouse models. *Front. Genet.* **5**, 88 (2014).

Point 5 raised by Reviewer #2:

A minor point – the symbols for statistical significance are inconsistent in the figures and figure legends.

Reply:

This is now changed; statistical significance is now consistent in all Figures, shown as *P ≤ 0.05, **P ≤ 0.01 and ***P ≤ 0.001. This is also stated in the statistical analysis section in Methods.

POINT-BY-POINT REPLY TO REVIEWER #3

After a short statement on the importance of our work, the Reviewer raised the following comments:

Point 1 raised by Reviewer #3:

The loss of TH-expressing neurons in the VTA appears to be well substantiated, although peculiar, because substantia nigra (SN) neurons are generally thought to be more vulnerable to cell death. The authors recognize this issue, but provide no plausible explanation as to why VTA neurons might be more vulnerable to the transgene in this model. In fact, they don't even show whether the transgene is expressed in the midbrain or selectively in VTA neurons.

Reply:

In agreement with the Reviewer's comment, we evaluated the APP^{swe} expression in 6 month-old mice (**new Supplementary Fig. 1a**), showing that the transgene is not selectively expressed only in VTA neurons but in all analyzed areas from Tg2576 mice, including the SNpc (VTA, SNpc, hippocampus, NAc core/shell and striatum). Thus, the selective death of VTA neurons cannot be explained by the restriction of transgene expression to this brain area.

We have now revised the Discussion section, reasoning that the selectivity in VTA DAergic degeneration in our model likely arises from a dying-back mechanism starting from the axonal terminal in projections areas (see also response to point 1-2 of Reviewer 1).

Point 2 raised by Reviewer #3:

The authors look at astrocyte activation in the VTA by staining for GFAP, but they should also measure microglia activation with Iba1 staining.

Reply:

In line with the Reviewer's suggestion, we measured microglia activation with Iba1 staining in 3-month-old animals. In agreement with DAergic cell death in the Tg2576 VTA, we observed a significant increase of Iba1 activation restricted to this area. The new data are shown in **Fig. 2c**.

Point 3 raised by Reviewer #3:

The authors show that there is less evoked DA release in the NAc shell but not NAc core, which is peculiar since both regions of the NAc are innervated preferentially by VTA neurons. The authors account for this difference by saying that the core receives input from the SN and provide 3 references (28-30). Two of the references are reviews that do not make this distinction and the third talks about role of shell and core but does not address the source of DA in those two regions. The authors need to provide a more compelling argument (or data) showing that DA in NAc core comes from the SN. There may well be sub-sets of VTA DA neurons, some of which project to core and some to shell, and those that project to the shell might be more vulnerable to transgene expression.

Reply:

We agree with the Reviewer and apologize for this error, which was based on earlier reports that did not differentiate well the boundaries between VTA and SNpc in the midbrain. The NAc core is indeed mainly innervated by the VTA and these neurons derive from an anatomically distinct VTA

subpopulation compared to neurons projecting to the NAc shell¹. We have now corrected the text accordingly, and added a part in the Discussion to argue that the discrepancy in our results regarding DA outflow in the core and shell might well derive from the heterogeneity in these VTA subpopulations that could account for a potential higher vulnerability of shell-projecting neurons.

1. Lammel, S. *et al.* Unique properties of mesoprefrontal neurons within a dual mesocorticolimbic dopamine system. *Neuron* **57**, 760–773 (2008).

Point 4 raised by Reviewer #3:

The authors claim that they cannot measure evoked DA release in the hippocampus and resort to microdialysis. However, they measure just as much DA by microdialysis in hippocampus (Fig. 4a) as they do in NAc (Fig. 3d); thus, they should also be able to measure evoked DA release.

*Most of the TH in the hippocampus comes from the locus ceruleus (LC), so it is not possible to conclude that the deficiency in hippocampal TH is due to loss of VTA neurons (Fig. 4c). There is a fairly persuasive paper indicating that most of the DA in the hippocampus actually comes from LC neurons (Smith, *J Neurosci* 32, 6072) so authors need to take that into account in their analysis and discussion.*

Reply:

We thank the Reviewer for his/her comment. The sensitivity of the amperometric electrode in measuring evoked DA release decreases rapidly in the presence of substantial basal DA, as it occurs in the case of the hippocampus. This is likely because DAT levels in the hippocampus are not as high as in the striatum or NAc¹, and basal DA is allowed to diffuse at greater distances, thus increasing the baseline noise. The situation is even more challenging in the mouse, where DA neurons are small and thus amperometric spikes are less detectable. These technical issues make amperometric recordings of evoked DA in the hippocampus less reliable, so we have chosen to avoid them. Of note, we used microdialysis probes of different membrane length in order to measure basal levels of DA in the hippocampus and NAc shell (bigger probes of 3 mm membrane length for the hippocampus and small probes of 1 mm for the NAc).

In line with the Reviewer's comment concerning the LC-TH⁺ projections to the hippocampus, we performed a stereological cell-count of TH⁺ cells in the LC of 6 month-old WT and Tg animals (**new Fig. 4c,d**). We did not observe any change in the number of LC-TH⁺ cells in Tg mice that could argue for degeneration in this area. However, since the majority of TH inputs to the hippocampus indeed come from the LC^{2,3} and older Tg mice are reported to show deficits in the LC^{4,5}, in the text we acknowledge that the reduced DA outflow in the hippocampus might be the combined result of VTA DA neuron degeneration and reduced axonal release from LC-TH⁺ neurons (even though the cell bodies in the LC still appear intact). This is in line with the hypothesis of a dying-back mechanism of TH⁺ terminals in AD, initiated in the projection areas affected by A β (see also response to point 1-2 of Reviewer 1).

1. Mennicken, F. *et al.* Autoradiographic localization of dopamine uptake sites in the rat brain with 3H-GBR 12935. *J. Neural. Transm. Gen. Sect.* **87**, 1–14 (1992).
2. Smith, C. C. & Greene, R. W. CNS dopamine transmission mediated by noradrenergic innervation. *J. Neurosci.* **32**, 6072–6080 (2012).
3. Takeuchi, T. *et al.* Locus coeruleus and dopaminergic consolidation of everyday memory. *Nature* **537**, 357–362 (2016).
4. Guérin, D. *et al.* Early locus coeruleus degeneration and olfactory dysfunctions in Tg2576 mice. *Neurobiol. Aging* **30**, 272–283 (2009).
5. Liu, Y. *et al.* Amyloid pathology is associated with progressive monoaminergic neurodegeneration in a transgenic mouse model of Alzheimer’s disease. *J. Neurosci.* **28**, 13805–13814 (2008).

Point 5 raised by Reviewer #3:

There seems to be some discrepancy in the analysis of hippocampal LTP which they say is normal at 2 and 3 months of age, even though there is already significant loss of VTA DA neurons at 3 months (Fig. 1).

Reply:

Cell loss in the VTA starts sometime between 2 and 3 months of age and LTP in 2 month-old Tg mice is normal. As now stated more clearly in the text, even though LTP at 3 months appears unchanged, hippocampal PSD composition is already compromised. Importantly, the hippocampal levels of D1 receptors in Tg mice are significantly increased, likely as a compensation to reduced levels of DA in the hippocampus; it is possible that at the early stages of cell death, this compensation is sufficient to maintain LTP at normal levels. In fact, we chose to perform the biochemical, electrophysiological and behavioral experiments in 6 month-old animals, when both VTA neuronal death and hippocampal synaptic plasticity impairments are well-established.

Point 6 raised by Reviewer #3:

The behavioral analysis is rather sparse. If the authors really want to examine reward learning they need to do more than a conditioned place preference experiment. Pavlovian and instrumental learning experiments along with progressive ratio experiments should be included. And, the authors need to examine whether the drug treatments restore normal behavior in their transgenic model.

Reply:

We thank the Reviewer for bringing this point to our attention. We agree with the Reviewer, particularly for the importance of Pavlovian learning. We decided to perform the place preference experiments because of the incentive learning nature of this task. The CPP is regarded as a Pavlovian paradigm as, for instance, reported in different sources: in *Neurobiology of Mental Illness* (Fourth Edition, Edited by DS Charney, EJ Nestler, P Sklar, and JD Buxbaum; Oxford University) “The CPP is a Pavlovian paradigm reflecting stimulus learning (...)”; in *Molecular Basis of Memory*, (1st Edition, Edited by EC Muly and Z Khan, Academic Press) “CPP paradigm is

a variant of passive Pavlovian conditioning where the animal is placed in a multichamber apparatus and learns to associate each chamber with a form of stimulation (...); in *Orexin/Hypocretin System*, (Edited by A Shekhar; Progress in Brain Research Elsevier) “The CPP paradigm measures Pavlovian approach toward an environment previously associated with reward”; and in *Behavioral Neuroscience of Drug Addiction* (by DW Self, JK Staley Gottschalk, Springer Science & Business Media, 2009) “...during training one context is consistently paired with the subjective experience of the drug, hence the Pavlovian aspect of this task.” In a *Nature Protocol* (1, 1662-70; 2006, Drug-induced conditioned place preference and aversion in mice by CL Cunningham, CM Gremel & PA Groblewski), place conditioning is defined as “a form of Pavlovian conditioning routinely used to measure the rewarding or aversive motivational effects of objects or experiences (e.g., abused drugs)”. Moreover, a recent opinion paper (What’s conditioned in conditioned place preference? By Huston et al., 2013; Trends in Pharmacological Sciences, Vol. 34, No. 3) has discussed that in the CPP exist “at least three major processes by which to account for why an animal spends more time in a place in which it received a reinforcer”. Besides the classical view of incentive-driven behavior also the operant conditioning of the behavior and the effect of conditioned treatment have been taken into account. These authors offer some points in support of the idea that CPP can also be viewed as operant conditioning task in which discriminative stimuli (proximal and distal) are associated via the reinforcer with the spontaneous behavior so that, during testing, the discriminative stimuli produce the reinforced behavior (that is “stay in this compartment”). In other words, they suggest that the measured behavior (place preference) is under control of the discriminative stimuli prevailing during the presentation of the reinforcer. Prudently, we may say that these are features shared with instrumental learning tasks.

Moreover, as suggested by the Reviewer, we performed an additional experiment testing the hypothesis that selegiline treatment could restore place conditioning in Tg2576 mice. We confirmed that selegiline treatment can abolish deficits in place preference and food consumption of Tg2576 mice. The new data are shown in **Fig. 7d,e**.

Point 7 raised by Reviewer #3:

Likewise, there are additional hippocampal dependent tasks should be included, e.g. Morris water maze. The contextual fear conditioning (CFC) data (Fig. 4f, 6e) are meaningless since the authors show that the control mice spent only 20 seconds 'freezing' after conditioning during a 5-min test period (p. 29 line 7). Most authors obtain 70-80% of time spent freezing with this paradigm.

Reply:

In line with the Reviewer's suggestion, we performed new experiments to evaluate the performance of Tg2576 mice during a Morris water maze task and we confirmed the ability of selegiline treatment to improve spatial memory deficits. The new data are reported in **Fig. 7c** and **Supplementary Fig. 5d-f**.

We agree with the Reviewer about the inconsistency of CFC results reported in the original version of the paper. In fact, regretfully, we did not explicate that time spent in freezing during context test was a mean calculated on the 5-min test period ("average minute" = total 5-min freezing/5). In the revised version of the manuscript we added this clarification in the Methods section, reporting total freezing time both for training and contest test. Consequently, we modified all Figures relevant to CFC data.

However, we would like to stress that the literature on fear conditioning in Tg2576 mice is rather sparse: besides authors who report impaired freezing elicited by contextual conditioning, other authors report aberrations in post-shock freezing during training¹, others report no deficits in conditioned fear elicited by contextual stimuli¹⁻³, or finally others report impaired freezing elicited only by tone conditioned stimulus^{1,3}. Further, many studies reporting CFC deficits show percentage of freezing inferior to 70-80%⁴⁻⁷. The differences in freezing time percentages might be ascribed to the different experimental protocols as well as to the different methods to measure freezing (e.g., automatic vs. manual scoring). In line with previous studies⁸⁻¹⁰, here we used a manual scoring of freezing, sensitive to detect even the finest movements.

1. Barnes, P. & Good, M. Impaired Pavlovian cued fear conditioning in Tg2576 mice expressing a human mutant amyloid precursor protein gene. *Behav. Brain Res.* **157**, 107–117 (2005).
2. Corcoran, K.A. *et al.* Overexpression of hAPP^{sw} Impairs Rewarded Alternation and Contextual Fear Conditioning in a Transgenic Mouse Model of Alzheimer's Disease. *Learn. Mem.* **9**, 243-252 (2002).
3. Lelos, M. J. & Good, M. A. β -Amyloid pathology alters neural network activation during retrieval of contextual fear memories in a mouse model of Alzheimer's disease. *Eur. J. Neurosci.* **39**, 1690–1703 (2014).
4. Imbimbo, B. P. *et al.* CHF5074, a novel gamma-secretase modulator, restores hippocampal neurogenesis potential and reverses contextual memory deficit in a transgenic mouse model of Alzheimer's disease. *J. Alzheimers Dis.* **20**, 159–173 (2010).
5. Perez-Cruz, C. *et al.* Reduced spine density in specific regions of CA1 pyramidal neurons in two transgenic mouse models of Alzheimer's disease. *J. Neurosci.* **31**, 3926–3934 (2011).
6. Rodriguez-Rivera, J. *et al.* Rosiglitazone Reversal of Tg2576 Cognitive Deficits is Independent of Peripheral Gluco-Regulatory Status. *Behav Brain Res.* **216**, 255-261 (2011).
7. Landlinger, C. *et al.* Active immunization against complement factor C5a: a new therapeutic approach for Alzheimer's disease. *J Neuroinflammation* **12**, 150 (2015).
8. Laricchiuta, D. *et al.* Effects of endocannabinoid and endovanilloid systems on aversive memory extinction. *Behav. Brain Res.* **256**, 101–107 (2013).
9. Cutuli, D. *et al.* n-3 polyunsaturated fatty acids supplementation enhances hippocampal functionality in aged mice. *Front. Aging Neurosci.* **6**, 220 (2014).
10. Laricchiuta, D. *et al.* Maintenance of aversive memories shown by fear extinction-impaired phenotypes is associated with increased activity in the amygdaloid-prefrontal circuit. *Sci. Rep.* **6**, 21205 (2016).

REVIEWERS' COMMENTS:

Reviewer #1 (Remarks to the Author):

I think the authors in this revised version of the paper answer all criticisms suggested by the reviewers. Now, the paper can be published in NC as it is. I think, the results presented in the paper are very important and could shed light on translational models of psychiatric disorders such as Alzheimer. Therefore, I strongly support its publication in Nature Communications.

Reviewer #2 (Remarks to the Author):

The authors have addressed my comments with additional experiments and data.

Reviewer #3 (Remarks to the Author):

This manuscript is considerably improved. The authors make a compelling case that an Alzheimer's mouse model affects dopamine signaling from the VTA, as well as hippocampal function and mouse behavior. Importantly, the authors can restore normal hippocampal function and behaviors with treatments that enhance dopamine levels.

There are a few issues that need to be addressed (without further review).

1. The summary on p. 4 lines 1-10 is redundant with Abstract and could be eliminated.
2. P5, line 9, The authors say twice that the TH-negative population is "constant and unchanged." The two words mean the same thing in this context and that fact is not a compelling argument that TH neurons are lost compared to the other data they present. I think it would be sufficient to simply point out that the TH-negative population is unchanged and then go on to describe their other data.
3. P11 line 8, I think the authors' mean that their treatment improves synaptic function rather than dysfunction.
4. P12, line 25, "localizatory" is not an English word.
5. P15 The authors say that segiline treatment restores DA in hippocampus (supplementary figure 4). Assuming that they only measured DA content in hippocampus, the conclusion that the behavioral rescue is due to enhanced DA in the hippocampus is not warranted. It could well be that restoration of DA in other or all targets of the VTA contribute to restoration of behaviors.
6. P14, line 16. I don't think there is a parabrachial nucleus of the VTA. The parabrachial nucleus that I know is near the cerebellum.
7. There is a new relevant paper linking dopamine action in hippocampus to locus ceruleus that the authors should consider. Locus coeruleus and dopaminergic consolidation of everyday memory. Takeuchi T, Duzskiewicz AJ, Sonneborn A, Spooner PA, Yamasaki M, Watanabe M, Smith CC, Fernández G, Deisseroth K, Greene RW, Morris RG. Nature. 2016 Sep 7;537(7620):357-362

POINT-BY-POINT REPLY TO REVIEWER #1

I think the authors in this revised version of the paper answer all criticisms suggested by the reviewers. Now, the paper can be published in NC as it is. I think, the results presented in the paper are very important and could shed light on translational models of psychiatric disorders such as Alzheimer. Therefore, I strongly support its publication in Nature Communications.

POINT-BY-POINT REPLY TO REVIEWER #2

The authors have addressed my comments with additional experiments and data.

POINT-BY-POINT REPLY TO REVIEWER #3

This manuscript is considerably improved. The authors make a compelling case that an Alzheimer's mouse model affects dopamine signaling from the VTA, as well as hippocampal function and mouse behavior. Importantly, the authors can restore normal hippocampal function and behaviors with treatments that enhance dopamine levels.

There are a few issues that need to be addressed (without further review).

1. The summary on p. 4 lines 1-10 is redundant with Abstract and could be eliminated.

Considering the Journal's format request to provide a brief summary of results in the last paragraph of the introduction, we chose to keep the text as it is. We would like to note that this paragraph contains a summary of additional data (recovery with L-Dopa and selegiline) to the ones described in the abstract (due to word limitations).

2. P5, line 9, The authors say twice that the TH-negative population is "constant and unchanged." The two words mean the same thing in this context and that fact is not a compelling argument that TH neurons are lost compared to the other data they present. I think it would be sufficient to simply point out that the TH-negative population is unchanged and then go on to describe their other data. This is now changed in the text to simply state that the TH-negative population is constant, and the repetition is omitted.

3. P11 line 8, *I think the authors' mean that their treatment improves synaptic function rather than dysfunction.*

The referee is correct, this is now changed in the text.

4. P12, line 25, *“localizatory” is not an English word.*

The word is now omitted.

5. P15 *The authors say that segiline treatment restores DA in hippocampus (supplementary figure 4). Assuming that they only measured DA content in hippocampus, the conclusion that the behavioral rescue is due to enhanced DA in the hippocampus is not warranted. It could well be that restoration of DA in other or all targets of the VTA contribute to restoration of behaviors.*

We now added a new phrase in the Discussion to indicate that the rescue of behavioral defects might co-involve other brain regions additionally to the hippocampus (P15 lines 1-3).

6. P14, line 16. *I don't think there is a parabrachial nucleus of the VTA. The parabrachial nucleus that I know is near the cerebellum.*

This is now changed to parabrachial pigmented area (PBP).

7. *There is a new relevant paper linking dopamine action in hippocampus to locus ceruleus that the authors should consider. Locus coeruleus and dopaminergic consolidation of everyday memory. Takeuchi T, Duzkiewicz AJ, Sonneborn A, Spooner PA, Yamasaki M, Watanabe M, Smith CC, Fernández G, Deisseroth K, Greene RW, Morris RG. Nature. 2016 Sep 7;537(7620):357-362*

This paper was already cited in the previous version (Ref. 42).